# Virus subtype-specific suppression of MAVS aggregation and activation by PB1-F2 protein of influenza A (H7N9) virus

**Pak-Hin Hinson Cheung**[1], **Tak-Wang Terence Lee**[1], **Chun Kew**[1], **Honglin Chen**[2], **Kwok-Yung Yuen**[2], **Chi-Ping Chan**[1]*, **Dong-Yan Jin**[1]*

**1** School of Biomedical Sciences, The University of Hong Kong, Pokfulam, Hong Kong, **2** State Key Laboratory for Emerging Infectious Diseases and Department of Microbiology, The University of Hong Kong, Pokfulam, Hong Kong

* chancp10@hku.hk (CPC); dyjin@hku.hk (DYJ)

## Abstract

Human infection with avian influenza A (H5N1) and (H7N9) viruses causes severe respiratory diseases. PB1-F2 protein is a critical virulence factor that suppresses early type I interferon response, but the mechanism of its action in relation to high pathogenicity is not well understood. Here we show that PB1-F2 protein of H7N9 virus is a particularly potent suppressor of antiviral signaling through formation of protein aggregates on mitochondria and inhibition of TRIM31-MAVS interaction, leading to prevention of K63-polyubiquitination and aggregation of MAVS. Unaggregated MAVS accumulated on fragmented mitochondria is prone to degradation by both proteasomal and lysosomal pathways. These properties are proprietary to PB1-F2 of H7N9 virus but not shared by its counterpart in WSN virus. A recombinant virus deficient of PB1-F2 of H7N9 induces more interferon β in infected cells. Our findings reveal a subtype-specific mechanism for destabilization of MAVS and suppression of interferon response by PB1-F2 of H7N9 virus.

## Author summary

Exactly why avian influenza A (H5N1) and (H7N9) viruses cause severe diseases in humans remains unclear. PB1-F2 protein encoded by influenza A virus is one virulence factor that might make a difference. In this study we show that PB1-F2 protein of H7N9 virus is particularly strong in the suppression of host antiviral defense. This was achieved by inhibiting a key protein in cell signaling named MAVS. PB1-F2 directs MAVS for degradation and prevents MAVS from forming protein aggregates required for full activation. A recombinant virus in which PB1-F2 of H7N9 has been deleted can activate host antiviral response robustly. Our findings reveal a novel mechanism by which PB1-F2 protein of H7N9 virus prevents MAVS aggregation and promotes MAVS degradation, leading to the suppression of host antiviral defense.

**Data Availability Statement:** All relevant data are within the manuscript and its Supporting Information files.

**Funding:** This work was supported by Hong Kong Health and Medical Research Fund (15140662, HKM-15-M01, 19180812 and 19181002). The funders had no role in study design, data collection and analysis, decision to publish, or preparation of the manuscript.

**Competing interests:** The authors have declared that no competing interests exist.

## Introduction

Avian influenza A (H5N1) and (H7N9) viruses occasionally cross species barrier to infect humans, causing severe respiratory disease and posing pandemic threats. In 1997, human infection with H5N1 virus in Hong Kong caused 6 deaths out of 18 infected individuals presented with viral pneumonia, acute respiratory distress syndrome and multiple organ failure [1, 2]. Since then, H5N1 has become a global epidemic with a case mortality of 55%, although the transmission route is restricted to avian-to-human contact [3]. In 2013, another novel avian subtype H7N9 emerged in humans in mainland China, presented with similar severe disease with 40% case mortality [4, 5]. At least four later H7N9 epidemics following the first wave in 2013 were recorded in mainland China from 2014 to 2017 with expanded geographical distribution and genetic diversity [6]. Although no evidence supports human-to-human transmission [7], studies using ferret model suggested H7N9 transmissibility between ferrets through respiratory droplets [8]. This is alarming and the pandemic potential of H7N9 and similar avian viruses in humans should not be underestimated. It remains elusive as to why H5N1 and H7N9 viruses are highly pathogenic in humans, but a robust innate immune response including a cytokine storm [4, 5], in which proinflammatory cytokines such as tumor necrosis factor (TNF) α are induced to very high levels for a long period of time [9], is thought to play an important role. Understanding the virulence factors of H5N1 and H7N9 viruses and how they dysregulate innate immunity might hold the key to the development of specific and effective therapeutics.

A small non-structural protein named PB1-F2 is one of the virulence factors of influenza A virus [10]. It is expressed from the largest +1 open reading frame of the PB1 segment of the viral genome, with a full length of 90 amino acids, but truncations and high sequence variability in this region have been found [11, 12]. PB1-F2 was initially identified as a mitochondrial protein that induces apoptosis of immune cells [13]. PB1-F2 deficiency results in attenuation of the virus in mammals, but host species- and virus subtype-specific effects of the loss of PB1-F2 have also been described [11, 12]. PB1-F2 is also known to interact and compete with HAX-1, an inhibitor of PA subunit of influenza A virus RNA polymerase [14, 15]. More importantly, PB1-F2 suppresses innate immunity and this suppression correlates with viral pathogenicity. PB1-F2 with an S66 residue has been found to be more pathogenic in mouse model than its mild counterpart with an N66 [16]. Mechanistically, S66 version of PB1-F2 suppresses RIG-I-MAVS signaling and early type I interferon (IFN) production [17] through dissipation of mitochondrial membrane potential (ΔΨm) and sequestration of MAVS adaptor protein [18–20]. More recently, PB1-F2 protein with an S66 pathogenic marker of the H1N1 subtype of the 1918 pandemic, which is also known to be highly pathogenic and probably derived from an avian virus, was found to specifically suppress TBK1-mediated type I IFN production by co-degradation with DDX3X [21]. H5N1 and H7N9 are highly pathogenic viruses that express full-length PB1-F2 without the S66 pathogenic marker [22, 23]. Since PB1-F2 sequences are known to be under a strong selection pressure [21], PB1-F2 proteins might have evolved to confer new features to enhance the pathogenicity of H5N1 and H7N9 viruses. This small protein can make a big difference as in the 1918 pandemic strain [21]. Whether and how type I IFN antagonism of PB1-F2 might contribute to high pathogenicity of H5N1 and H7N9 remain to be elucidated.

In this study, we sought to evaluate the effect of H7N9 PB1-F2 on innate IFN production. H7N9 PB1-F2 was found to form extensive protein aggregates on mitochondria to perturb innate immune signal propagation through MAVS. Generally consistent with previous reports [24–27], H5N1 PB1-F2, but not WSN PB1-F2, had similar aggregation propensity. H7N9 PB1-F2 exerted an inhibitory effect on MAVS aggregation by preventing TRIM31 ubiquitin

ligase from interacting with MAVS and thus reducing K63-polyubiquitination of MAVS. This accelerated proteasomal and lysosomal degradation of MAVS leading to inhibition of host antiviral response against H7N9 virus. This virus subtype-specific suppression of MAVS aggregation and type I IFN production by H7N9 PB1-F2 has implications in H7N9 pathogenesis and disease intervention.

## Results

### Unique features of H7N9 PB1-F2 protein

H7N9 and H5N1 viruses are known to be more virulent than human influenza viruses including the WSN laboratory strain [28]. When we compared amino acid sequences of PB1-F2 proteins of H7N9, H5N1 and WSN viruses (Fig 1A), 60% identity was found between WSN and the other two strains. Consistent with their closer evolutionary relationship, PB1-F2 proteins of H5N1 and H7N9 shared 82.2% identical residues. Notably, although H5N1 and H7N9 viruses were highly virulent in humans, their PB1-F2 proteins carried N66 (Fig 1A) but not the previously described pathogenic marker S66 as seen in the 1918 pandemic strain [16].

PB1-F2 protein is prone to proteasomal degradation [13, 21]. In keeping with this, when it was expressed in mammalian HEK293T cells, Flag-tagged PB1-F2 (PB1-F2-Flag) of WSN strain was barely detected but its steady-state level was much increased upon treatment of cells with proteasome inhibitor MG132 (Fig 1B, upper panel, lane 2 vs 1). In contrast, H7N9 PB1-F2 was most abundant in HEK293T cells in the absence of MG132 (Fig 1B, upper panel, lane 5 vs 1 and 3) and its protein level was minimally affected by MG132 treatment (lane 6 vs 5). Interestingly, the expression level of H5N1 PB1-F2 in HEK293T cells in the absence of MG132 was between those of its counterparts in WSN and H7N9 (Fig 1B, upper panel, lane 3 vs 1 and 5), and it was slightly elevated when cells were treated with MG132 (lane 4 vs 3). Similar results were also observed in chicken DF-1 cells (Fig 1B, lower panel). These results might indicate the unique features of H7N9 PB1-F2 in protein expression, abundance, stability and susceptibility to proteasomal degradation.

### Aggregation of H7N9 PB1-F2

WSN PB1-F2 has been shown to adopt β-sheet conformation for amyloid aggregation in a membranous environment [24–27]. Particularly, antiparallel cross-β-sheet pairing is necessary for the formation of prefibrillar oligomer that seeds amyloid fibrils [30–32]. To explore whether H7N9 PB1-F2 might form similar conformation seen in WSN PB1-F2, the PASTA 2.0 program was used to predict cross-β sheet structure of PB1-F2 sequences [29]. A threshold energy was set at -2.8 PEU, below which candidate pairs represent highly confident cross-β interaction with 90% specificity [29]. While PB1-F2 proteins from WSN, H5N1 and H7N9 viruses might adopt parallel pairing at residues 55–58 with an energy of lower than -5.31 PEU, multiple highly probable anti-parallel pairings were also predicted to form in PB1-F2 of H5N1 and H7N9 (Fig 1C and S1 Fig). For H7N9 PB1-F2, 12 putative anti-parallel pairings with a low energy ranging from -3.26 to -3.97 PEU were predicted to span the entire C terminus (Fig 1C and S1 Fig). Likewise, H5N1 PB1-F2 might form 12 anti-parallel pairings with an energy ranging from -2.98 to -3.53 PEU at its C-terminal region. In contrast, WSN PB1-F2 had only 6 putative anti-parallel pairings with the highest energy ranging from -2.88 to -3.21 PEU in the same region.

Stronger anti-parallel cross-β-sheet pairing in PB1-F2 from H5N1 and H7N9 might stabilize their prefibrillar oligomers leading to higher aggregation propensity. To check for this possibility, SDD-AGE and protein solubility assays were performed to compare the aggregation propensity of PB1-F2 proteins from WSN, H5N1 and H7N9. SDD-AGE is a non-denaturing

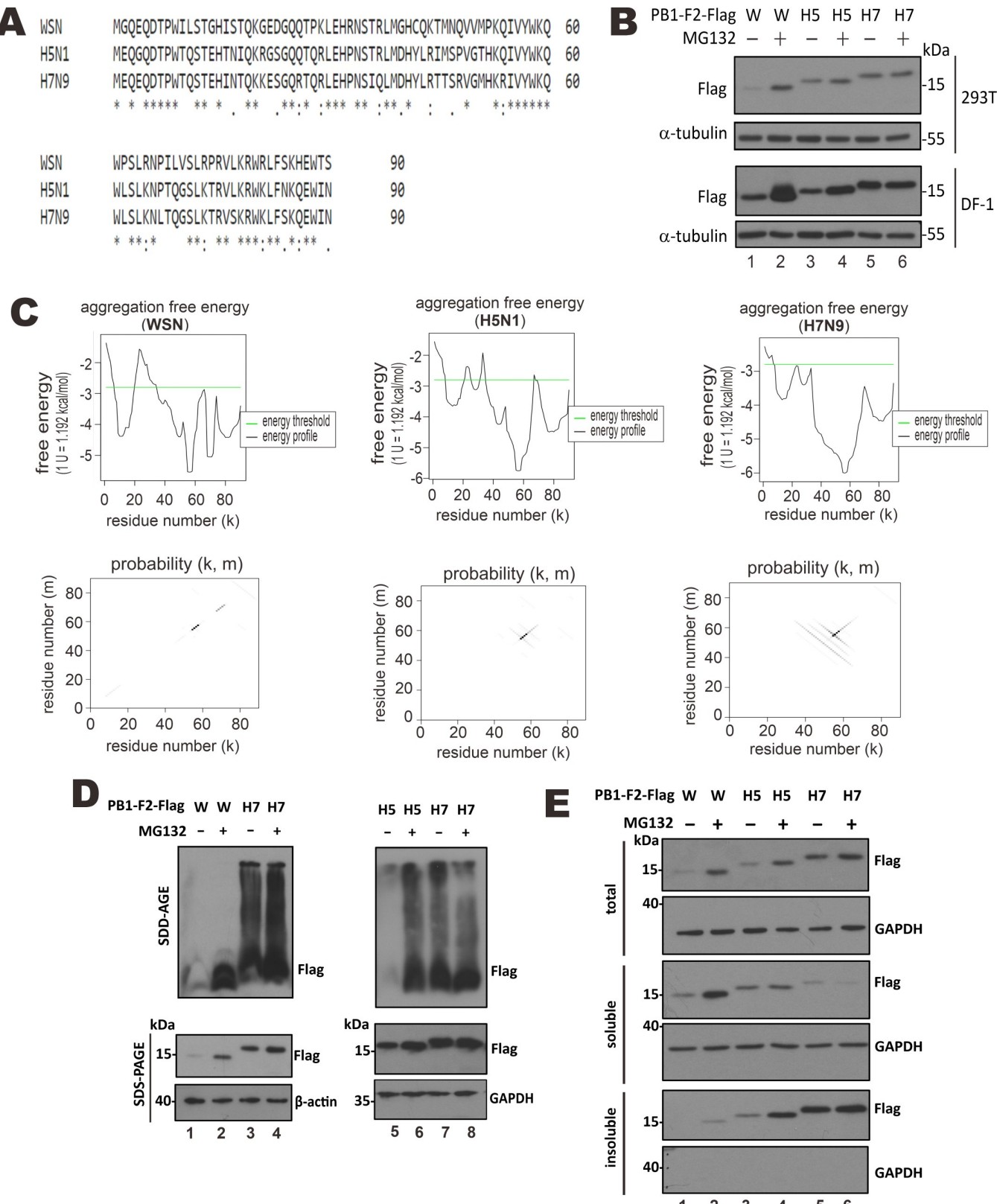

**Fig 1. High aggregation potential of H7N9 PB1-F2.** (A) Sequence alignment of PB1-F2 proteins from WSN, H5N1 and H7N9 viruses. Amino acid sequences of PB1-F2 proteins from human WSN, H5N1 and H7N9 viruses were aligned by the Clustal Omega program (https://www.ebi.ac.uk/Tools/msa/clustalo/).

Identical, similar and weakly similar residues are indicated by *,: and., respectively. (**B**) Protein expression. HEK293T and DF-1 cells in 6-well plates were transfected with expression construct (1 μg) for PB1-F2-Flag from WSN (W), H5N1 (H5) and H7N9 (H7) viruses. After 42 hours, cells were either mock treated with DMSO or treated with 10 μM MG132 for 6 more hours and then subjected to total protein extraction and Western blot analysis with anti-Flag. α-tubulin served as an internal control for protein loading. Slight differences in the actual molecular masses of PB1-F2 proteins from WSN, H5N1 and H7N9 viruses on the 12% polyacrylamide gel might be explained at least partially by the differences in their calculated molecular masses. (**C**) Bioinformatic prediction of cross-β sheet structure. Amino acid sequences of PB1-F2 from WSN, H5N1 and H7N9 viruses were analyzed for putative cross-β sheet structure by PASTA 2.0 (http://protein.bio.unipd.it/pasta2/). The top panels were the free-energy profile plotting free energy of cross β sheet pairing against residue number k. The bottom panels represented probability matrix that showed probability of pairing of residues m and k of two self-aligned PB1-F2 amino acid sequences. Probability calculation was as described [29]. Dot intensity was proportional to probability of pairing. (**D and E**) Aggregation and solubility assays. HEK293T cells in 6-well plates were transfected with PB1-F2-Flag expression construct (1 μg). After 42 hours, cells were either mock treated with DMSO or treated with 10 μM MG132 for 6 more hours and subjected to SDD-AGE and SDS-PAGE analysis (D) as well as protein solubility assay with RIPA lysis buffer (E). GAPDH or β-actin served as internal control for protein loading. Similar results were obtained from three independent experiments.

method to identify high-molecular-mass protein complex such as prion or amyloid aggregates [33–35]. MG132 was used to restore the expression of WSN PB1-F2 and H5N1 PB1-F2. Generally consistent with Fig 1B, WSN PB1-F2 and H5N1 PB1-F2 were much stabilized by MG132 (Fig 1D, lanes 2 and 6). In contrast, the stabilizing effect of MG132 on H7N9 PB1-F2 was modest at best (Fig 1D, lanes 4 and 8). WSN PB1-F2 remained to be low-molecular-mass species with and without MG132 treatment on the SDD-AGE blot (Fig 1D, lanes 1 and 2). Notably, high-molecular-mass species of H7N9 PB1-F2 appeared on the SDD-AGE blot (Fig 1D, lanes 3, 4, 7 and 8). Although H5N1 PB1-F2 was less abundant than H7N9 PB1-F2, high-molecular-mass species of H5N1 PB1-F2 were also seen upon MG132 treatment (Fig 1D, lane 6). In contrast, high-molecular-mass species of H7N9 PB1-F2 were visible in the absence of MG132 and the potentiating effect of MG132 on them was less evident (Fig 1D, lanes 4 and 8). These results were consistent with the prediction that PB1-F2 of H5N1 and H7N9 had higher aggregation propensity than the counterpart in WSN. To verify this, protein solubility assay was performed in RIPA lysis buffer with 0.1% SDS to assess the aggregation propensity of PB1-F2 proteins. Whereas WSN PB1-F2 was highly soluble in RIPA lysis buffer no matter cells were treated with MG132 or not (Fig 1E, lanes 1 and 2), H7N9 PB1-F2 was sufficiently insoluble (lanes 5 and 6). H5N1 PB1-F2 was found in both soluble and insoluble fractions, but it became more insoluble when its expression was restored by MG132 (Fig 1E, lanes 3 and 4). In short, PB1-F2 proteins of H7N9 and H5N1 might have higher propensity to form antiparallel cross-β sheet pairing and high-molecular-mass aggregates than WSN PB1-F2.

## Co-localization of H7N9 PB1-F2 with mitochondrial clusters

PB1-F2 is known to localize to mitochondria where it dissipates ΔΨm. However, PB1-F2 of A/PR8/1934/H1N1 (PR8) strain with 87 amino acid residues was extensively used for this type of studies [13, 20, 36, 37]. Some analysis was also performed with PB1-F2 proteins from other strains including H5N1 [38, 39]. To clarify whether this property of PB1-F2 has subtype or strain specificity, confocal staining was performed to test for mitochondrial localization and ΔΨm-dissipating activity of PB1-F2 proteins of PR8, WSN, H5N1 and H7N9 viruses. A cation dye specific for mitochondria named Mitotracker Red CMXRos was used for costaining. Notably, the fluorescent intensity of this dye is proportional to ΔΨm [40]. A significant reduction of MitoTracker staining was only observed when PR8 PB1-F2 was expressed (Fig 2A). Mitotracker staining remained unchanged in cells expressing PB1-F2 of any of the other three strains (Fig 2A). To confirm this result, JC-1 dye, which can distinguish polarized mitochondria by a high red-to-green emission ratio [41], was used to measure ΔΨm of PB1-F2-expressing cells. We found that the red-to-green ratio of JC-1 was reduced only when cells were treated with carbonyl cyanide m-chlorophenyl hydrazone (CCCP), an uncoupler of mitochondrial oxidative phosphorylation, or when PR8 PB1-F2 was expressed (Fig 2B, bars 2 and 3 vs

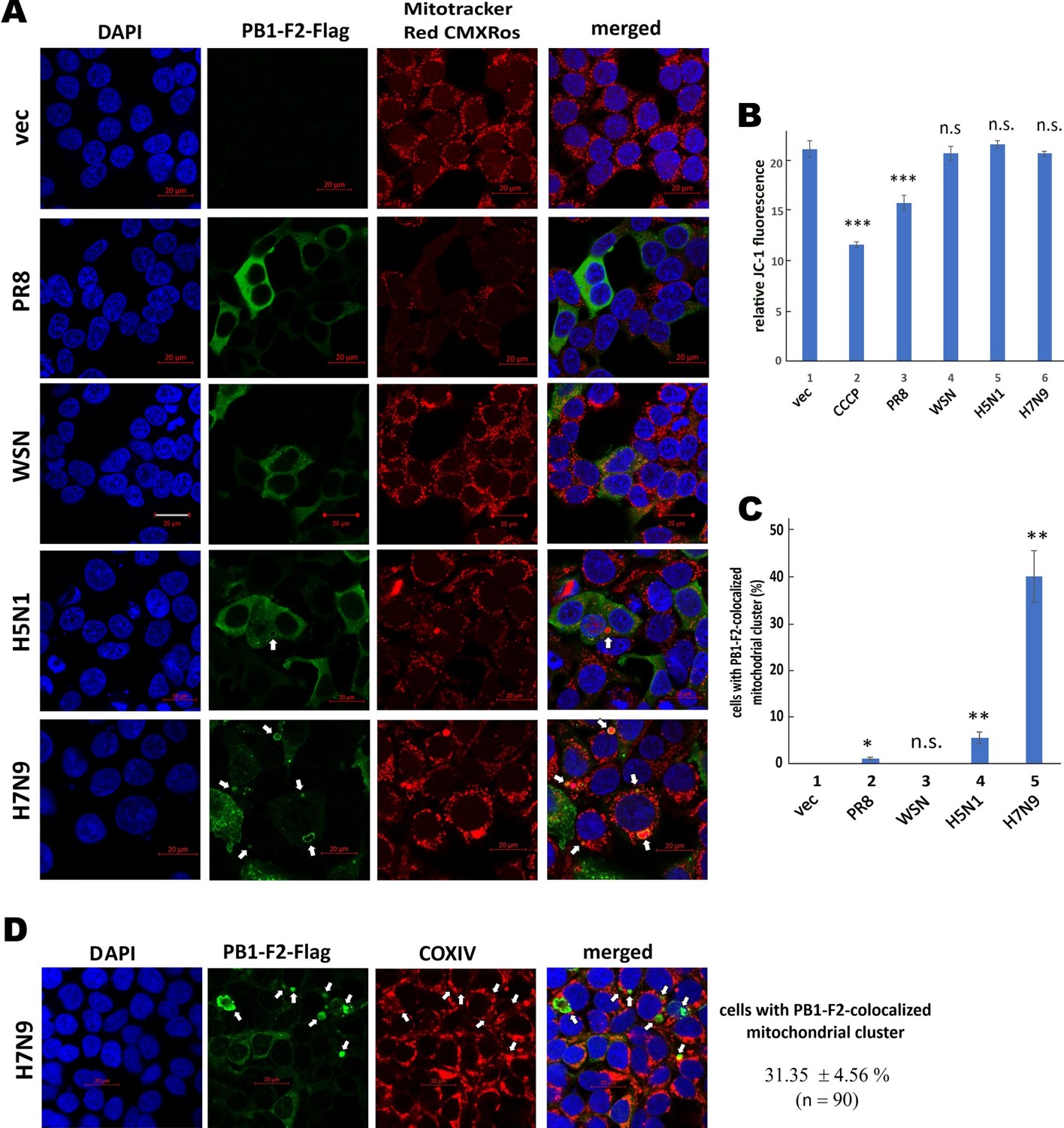

**Fig 2. Influence of H7N9 PB1-F2 expression on mitochondrial morphology and function.** (**A**) Mitotracker Red CMXRos staining. HEK293T cells over coverslips in 6-well plates were transfected with 1μg of PB1-F2-Flag expression construct. After 48 hours, cells were stained with 500 nM Mitotracker Red CMXRos for 30 min and then fixed with 4% paraformaldehyde. PB1-F2 was probed with anti-Flag and nuclei were stained with DAPI. The stained cells were analyzed by confocal microscopy. Arrows indicated distinct mitochondrial clusters colocalized with PB1-F2. vec: vector control. Bars, 20 μm. (**B**) JC-1 staining. HEK293T cells in 12-well plates were transfected with 0.5 μg of PB1-F2-Flag expression construct. After 48 hours, cells were stained with JC-1 dye. For CCCP control, cells were co-treated with 50 μM CCCP and JC-1 dye. JC-1 dye-stained cells were analyzed through fluorescence microplate reader for red and green signals, which respectively indicated healthy and

depolarized mitochondria. The data represent mean values ± SD of red-to-green ratio of JC-1 staining of three independent experiments. Unpaired Student's t-test was used to assess the statistical significance of the difference between vector control (vec) and sample. ***: P < 0.001. **: P < 0.01. *: P < 0.05. n.s.: P > 0.05. (**C**) Mitochondrial clustering. Cells with clustered mitochondria colocalized with PB1-F2 as indicated by arrows in (A) were calculated by number of cells with clustered mitochondria over total number of cells. For each experiment, at least 90 total cells were counted. The data represent mean values ± SD of cells (%) from three independent experiments. (**D**) COXIV staining. HEK293T cells over coverslips in 6-well plates were transfected with 1 μg of expression construct for Flag-tagged H7N9 PB1-F2. After 48 hours, cells were fixed with 1:1 methanol: acetone solution and probed with anti-Flag for PB1-F2, anti-COXIV for mitochondria and DAPI for nuclei. The stained cells were analyzed by confocal microscopy. Cells with clustered mitochondria colocalized with PB1-F2 were calculated as in (C).

1). Expression of PB1-F2 from WSN, H5N1 and H7N9 did not affect JC-1 dye staining. In addition, we observed that H7N9 PB1-F2 was concentrated to mitochondrial clusters as indicated by arrows (Fig 2A). These clusters are likely derived from mitochondrial fission [20, 42]. Co-localization of H7N9 PB1-F2 and mitochondrial clusters were further confirmed by Z-stack imaging at various focal planes through Z-axis spaced by 0.7 μm (S2 Fig). The mitochondrial clustering pattern was less prominent upon expression of H5N1 PB1-F2 (Fig 2A). On the contrary, the staining of PR8 PB1-F2 and WSN PB1-F2 was diffusive or punctate in the cytoplasm and mitochondria (Fig 2A). Quantitatively, mitochondrial clusters colocalized with PB1-F2 were seen in about 40% of H7N9 PB1-F2-expressing cells and 5% for H5N1 PB1-F2-expressing cells (Fig 2C). They were observed in only 0.9% of PR8 PB1-F2-expressing cells and were absent in cells expressing WSN PB1-F2 or receiving vector control (Fig 2C). Another mitochondrial marker cytochrome C oxidase subunit IV (COXIV) was also found to co-localize with H7N9 PB1-F2 showing mitochondrial clustering pattern in about 30% of expressing cells (Fig 2D). Thus, H7N9 PB1-F2, and H5N1 PB1-F2 to a lesser extent, did not decrease ΔΨm but colocalized with mitochondrial clusters.

## Suppression of MAVS-dependent type I IFN production by H7N9 PB1-F2

Mitochondrial adaptor protein FAF1 has recently been found to form aggregates that suppress RIG-I signaling [43]. As demonstrated above, PB1-F2 proteins of H5N1 and H7N9 did not decrease ΔΨm but formed protein aggregates on mitochondria. This prompted us to ask whether they might share RIG-I-suppressing activity similar to that of FAF1. WSN PB1-F2 and H5N1 PB1-F2 have also been shown to suppress type I IFN production [17, 44]. To shed light on the IFN antagonism of H7N9 PB1-F2, luciferase reporter assays were performed as previously described [45]. HEK293T cells expressing PB1-F2 of WSN, H5N1 or H7N9 and IFNβ-Luc reporter were either infected with Sendai virus or transfected with poly(I:C). Whereas expression of WSN PB1-F2 minimally affected Sendai virus- or poly(I:C)-induced activation of IFNβ promoter activity (Fig 3A and 3B, bar 3 vs 2), this activity was dampened by 50% when H5N1 PB1-F2 or H7N9 PB1-F2 was expressed (bars 4 and 5 vs 2). Likewise, in contrast to a mild suppressive effect of WSN PB1-F2 on the IFNβ-inducing activity of RIG-IN, a dominant active version of RIG-I [46], the suppression of RIG-IN activity by H5N1 PB1-F2 and H7N9 PB1-F2 was more pronounced (Fig 3C, bars 4 and 5 vs 2). The specificity of this suppression on RIG-I signaling was verified since PB1-F2 from any of the three viruses had no effect on MyD88-induced activation of IFNβ promoter activity (Fig 3D, bars 3–5). To shed light on the mechanism of action of PB1-F2, IFNβ promoter was activated by three downstream effector proteins of RIG-I signaling, namely MAVS (Fig 3E), TBK1 (Fig 3F) and a constitutively active form of IRF3 known as IRF3-5D [47] (Fig 3G). All three PB1-F2 proteins suppressed the activity of MAVS (Fig 3E, bars 3–5) but had no influence on TBK1 or IRF3-5D (Fig 3F and 3G), consistent with an action point upstream of TBK1. The suppressive activity of H7N9 PB1-F2 on MAVS was most prominent followed by that of H5N1 PB1-F2 (Fig 3E, bars 4 and 5 vs 2). The effect of WSN PB1-F2 was milder (Fig 3E, bar 3 vs 2). A similar suppressive pattern on MAVS activity was also observed when luciferase reporter expression was driven by

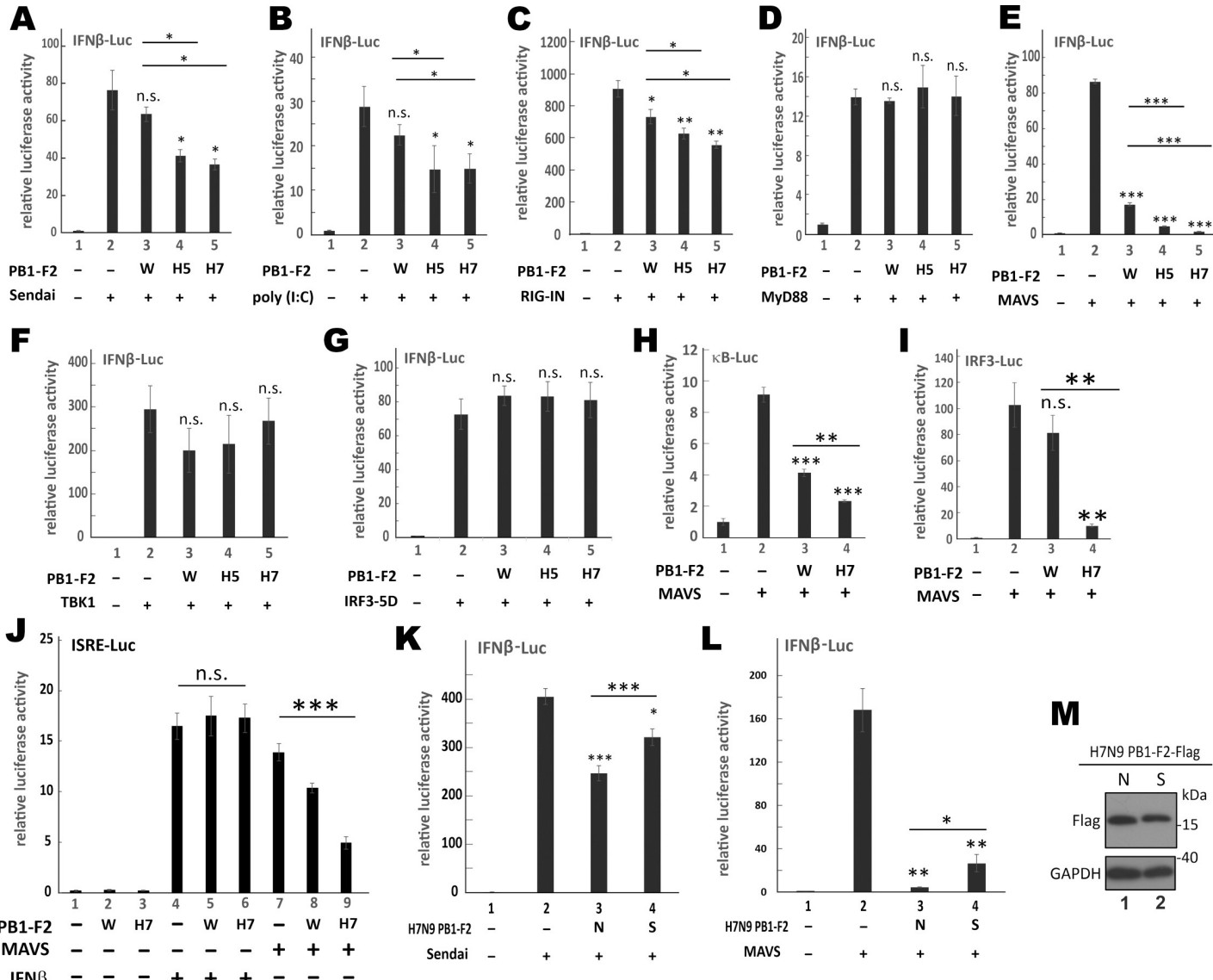

**Fig 3. Suppression of RIG-I-MAVS antiviral signaling by H7N9 PB1-F2. (A and B)** HEK293T cells in 12-well plates were transfected with 400 ng PB1-F2 expression constructs, 200 ng p125-Luc and 10 ng pRL-TK. After 24 hours, cells were treated with 100 hemagglutinating units/ml of Sendai virus (A) or 1 µg/mL of poly (I:C) (B) for a further 16 hours. Then, cells were harvested for dual-luciferase reporter assay. **(C-G)** HEK293T cells in 24-well plates were transfected with 200 ng PB1-F2 expression constructs, 100ng p125-Luc, 10ng pRL-TK and 50ng expression constructs for type I IFN stimulants including RIG-IN (C), MyD88 (D), MAVS in pEF-Bos vector (E), TBK1 (F) or IRF3-5D (G). After 48 hours, cells were harvested for dual-luciferase reporter assay. **(H and I)** HEK293T cells in 24-well plates were transfected with 200 ng PB1-F2 expression constructs, 100 ng kB-FLuc (H) or IRF3-FLuc (I), 10 ng pRL-TK with or without 50 ng MAVS expression construct for 48 hours before harvested for dual-luciferase reporter assay. **(J)** HEK293T cells in 24-well plates were transfected with 200 ng PB1-F2 expression constructs, 50 ng MAVS expression construct, 10 ng pRL-TK plus 100ng ISRE-FLuc. After 24 hours, cells were mock-treated or treated with 1000 U/mL human recombinant IFNβ protein for 24 hours before harvested for dual-luciferase reporter assay. **(K)** Same as in (A), except the inclusion of a group for H7N9 PB1-F2 S66 (S). **(L)** Same as in (E), except that 50 ng H7N9 PB1-F2 N66 (N) or S66 (S) was used. **(M)** HEK293T cells in 6-well plates were transfected with 1 µg expression construct for H7N9 PB1-F2 with N66 (N) or S66 (S) for 48 hours before SDS-PAGE and Western blot analysis. Anti-Flag recognized PB1-F2 protein. GAPDH served as internal loading control. All bars denote means ± SD of triplicate experiments. Unpaired Student's t-test was performed to evaluate the statistical significance of the difference between vector control (vec) and the indicated sample. ***: P < 0.001. **: P < 0.01. *: P < 0.05. n.s.: P > 0.05.

other three types of enhancer elements, namely κB elements (Fig 3H), IRF3 elements (Fig 3I) and IFN-stimulated response elements (ISRE; Fig 3J). These results indicated that H7N9 PB1-F2 and H5N1 PB1-F2 had a potent suppressive effect on MAVS. To verify the expression

of PB1-F2s and other proteins of interest in the luciferase assay, Western blot analysis was performed (S3 Fig). As also shown in Figs 1 and 2, H5N1 PB1-F2 and H7N9 PB1-F2 were more abundant than WSN PB1-F2, although the same dose of plasmid was used in the transfection (S3A–S3H Fig). Expression levels of RIG-IN (S3C Fig), MyD88 (S3D Fig), TBK1 (S3F Fig) and IRF3-5D (S3G Fig) were comparable in the presence and absence of PB1-F2s. However, MAVS expression was diminished when PB1-F2 was present. Notably, the diminishing effect of H5N1 PB1-F2 and H7N9 PB1-F2 was most prominent (S3E and S3H Fig).

PB1-F2 proteins of PR8 and H5N1 carrying the S66 pathogenic marker were found to suppress type I IFN production through MAVS [17–19, 48]. To test if S66 might be influential on the suppressive effect on MAVS, an N66S mutation was introduced to H7N9 PB1-F2. Surprisingly, the suppression of Sendai virus- or MAVS-induced activation of IFNβ promoter activity by H7N9 PB1-F2 N66 was more pronounced than that mediated by H7N9 PB1-F2 S66 (Fig 3K and 3L). The levels of both proteins were comparable when expressed alone (Fig 3M), but the steady-state amount of H7N9 PB1-F2 S66 was reduced than that of H7N9 PB1-F2 N66 upon stimulation with Sendai virus (S3I Fig) or co-expression of MAVS (S3J Fig). The diminishing effect of H7N9 PB1-F2 N66 on MAVS was more pronounced than that of H7N9 PB1-F2 S66 mutant (S3J Fig). The steady-state levels of PB1-F2 proteins of PR8 and H5N1 with S66 were lower in Sendai virus-infected cells, but much more pronounced IFN suppressive and pathogenic effects of PB1-F2 S66 were observed compared to those of PB1-F2 N66 [17–19, 48]. In contrast, H7N9 PB1-F2 S66 did not show an enhanced IFN suppressive effect. These results suggested that H7N9 PB1-F2 likely suppressed MAVS through another S66-independent mechanism.

## H7N9 PB1-F2 suppresses type I IFN production during influenza A virus infection

To monitor if PB1-F2 has an impact on type I IFN response during influenza A virus infection, recombinant viruses H7-2 wild type (WT) and H7-2 ΔF were made. These reassortant viruses contain all segments of H7N9 except that the segments expressing HA and NA are from WSN. They can therefore be handled in the regular Biosafety Level 2 facility. Whereas PB1- and PB1-F2-coding regions were intact in H7-2 WT, PB1-F2-coding sequence in H7-2 ΔF was completely disrupted by mutating start codon and inserting two premature stop codons without changing PB1-coding sequence (Fig 4A). These viruses were used to infect two model cell lines THP-1 and A549 at a multiplicity of infection (MOI) of 1. IFNβ mRNA expression was less robust in THP-1 cells infected with H7-2 WT than in cells infected with H7-2ΔF at 12 and 24 hours-post-infection (hpi; Fig 4B). Levels of viral transcripts (represented by those of segments 4 and 6) and proinflammatory cytokine mRNAs (represented by that of TNFα) were not significantly different between cells infected with H7-2 WT and H7-2 ΔF (Fig 4B), suggesting that H7N9 PB1-F2 specifically suppressed the production of type I IFNs but not proinflammatory cytokines. The replication of H7-2 WT in A549 cells was more robust than that of H7-2ΔF (Fig 4C). This was generally consistent with previous reports that PB1-F2 proteins of H9N2, H5N1 and H7N9 promote viral replication in human lung epithelial cells (i.e. A549) by preventing HAX-1 from inhibiting viral polymerase subunit PA [14, 15]. However, although enhanced replication of H7-2 WT virus resulted in stronger induction of TNFα mRNA at 6, 12 and 24 hpi, production of IFNβ mRNA was not enhanced at 12 hpi and even suppressed at 24 hpi (Fig 4C). This was compatible with the suppression of IFNβ expression by H7N9 PB1-F2 in infected A549 and THP-1 cells.

As mentioned above, H7N9 PB1-F2 was found to be a more potent suppressor of MAVS-mediated IFN response than WSN PB1-F2 (Fig 3A to 3J). To verify this in the context of viral

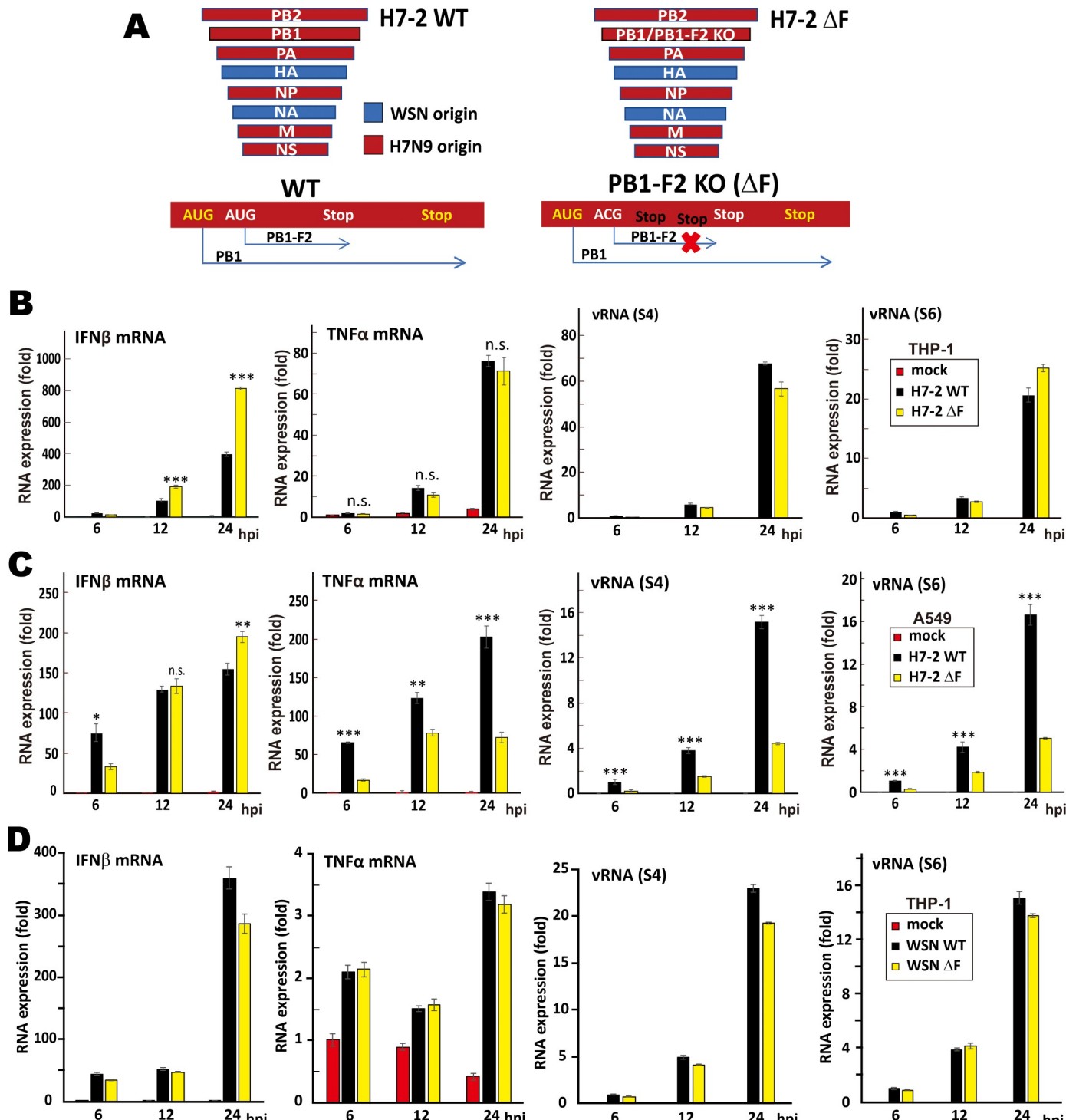

**Fig 4. Suppression of type I IFN response by H7N9 PB1-F2 in cells infected with recombinant influenza A virus.** (**A**) Schematic representation of the genotypes of H7-2 WT and H7-2 ΔF viruses. (**B and C**) THP-1 (B) and A549 (C) cells were infected with H7-2 WT and H7-2ΔF at MOI = 1. At 6, 12, and 24 hpi, infected cells were harvested for RT-qPCR assay for IFNβ mRNA, TNFα mRNA, vRNA segment 4 (S4) and segment 6 (S6). The levels of mRNA or vRNA relative to GAPDH mRNA were analyzed by using comparative Ct method. Bars represent means ± SD of triplicate experiments. Unpaired Student's t-test was performed to judge the statistical significance of the difference between H7-2 WT and H7-2ΔF groups. ***: P < 0.001. **: P < 0.01. *: P < 0.05. n.s.: P > 0.05. (**D**) WSN PB1-F2 does not suppress type I IFN response during viral infection. THP-1 cells were infected with WSN WT or WSN ΔF at MOI = 1. At 6, 12, and 24 hpi, infected THP-1 cells were harvested for RT-qPCR assay for IFNβ mRNA, TNFα mRNA, vRNA segment 4 (S4) and segment 6 (S6). The levels of mRNA or vRNA relative to GAPDH mRNA were analyzed by using comparative Ct method. Bars represent means ± SD of triplicate experiments.

infection, WSN virus (WSN WT) and its PB1-F2 defective mutant (WSN ΔF) were also constructed. THP-1 cells were then infected with WSN WT and WSN ΔF. Levels of IFNβ mRNA, TNFα mRNA and viral RNA segments were compared (Fig 4D). Unlike H7-2 WT and H7-2 ΔF, WSN WT and WSN ΔF did not show remarkable difference in either replication kinetic or induction of IFNβ and TNFα transcripts in THP-1 cells (Fig 4D). Hence, WSN PB1-F2 had no influence on type I IFN induction during viral infection in THP-1 cells, whereas H7N9 PB1-F2 exerted a significant IFN suppressive effect.

## H7N9 PB1-F2 destabilizes MAVS protein when RIG-I signaling is activated

Upon activation of RIG-I signaling, MAVS protein is strictly regulated by both proteasomal and lysosomal degradation to prevent uncontrolled signal activation [49]. We monitored MAVS expression levels in H7-2 WT- and H7-2 ΔF-infected THP-1 cells over a time course. Decrease in the protein level of MAVS was seen in H7-2 WT-infected cells at as early as 6 hpi, but the drop was noticeable in H7-2 ΔF-infected cells only at 12 hpi (Fig 5A, lane 2 vs 1 and 3, and lane 6 vs 3). This difference could not be accounted for by the difference in MAVS mRNA levels (Fig 5B). Viral protein expression as represented by the level of PA protein did not differ between cells infected with H7-2 WT and H7-2 ΔF (Fig 5A). H7N9 PB1-F2 was found to express early at 6 and 12 hpi, and it was not detectable at 24 hpi (Fig 5A). The same infection experiment was performed in THP-1 cells with WSN WT and WSN ΔF to verify whether the diminishing effect of PB1-F2 on MAVS protein might be virus type-specific (Fig 5C). We noted that the levels of MAVS protein were similar between cells infected with WSN WT and WSN ΔF (Fig 5C). A drop in MAVS mRNA level was seen in infected cells at as early as 6 hpi, but no difference was noticeable between cells infected by the two viruses (Fig 5D). This indicated a PB1-F2-mediated decrease in the level of MAVS protein in the early phase of infection with H7N9 but not WSN. This finding was compatible with destabilization of MAVS protein by H7N9 PB1-F2. To test this possibility, HEK293T cells overexpressing MAVS and PB1-F2 through co-transfection were treated with cycloheximide to terminate protein translation so that MAVS protein level was only affected by degradation. H7N9 PB1-F2 but not WSN PB1-F2 reduced the half-life of MAVS protein (Fig 5E, lanes 9–12 vs 5–8). Consistent with this, expression of H7N9 PB1-F2 diminished the levels of MAVS protein in a dose-dependent manner (Fig 5F, lanes 7–11). This effect was not observed for WSN PB1-F2 even at the highest expression level (Fig 5F, lane 6).

Interestingly, H7N9 PB1-F2 was not influential on the steady-state level of endogenous MAVS protein under mock-stimulated conditions (Fig 5G and 5I). However, upon transfection with poly (I:C) or infection with Sendai virus, protein expression levels of MAVS were diminished when H7N9 PB1-F2, but not WSN PB1-F2, was expressed (Fig 5G and 5I), whereas mRNA levels of MAVS remained unchanged (Fig 5H and 5J). We also observed that the destabilizing effect of H5N1 PB1-F2 on MAVS was similar to that of H7N9 PB1-F2 (Fig 5G anf 5I). In short, PB1-F2 proteins of H7N9 and H5N1 but not the counterpart in WSN enhanced MAVS degradation in immune-stimulated cells.

## H7N9 PB1-F2 promotes proteasomal and lysosomal degradation of aggregated MAVS

H7N9 PB1-F2 destabilized MAVS protein to its basal level when cells were infected with Sendai virus or transfected with poly(I:C) or MAVS expression vector (Fig 5). It was noteworthy that under all these immune-stimulatory conditions, self-aggregation of MAVS protein is also promoted through its CARD domain to form giant protein filament on mitochondria, which is required for propagation of type I IFN response [50–52]. To explore how H7N9 PB1-F2

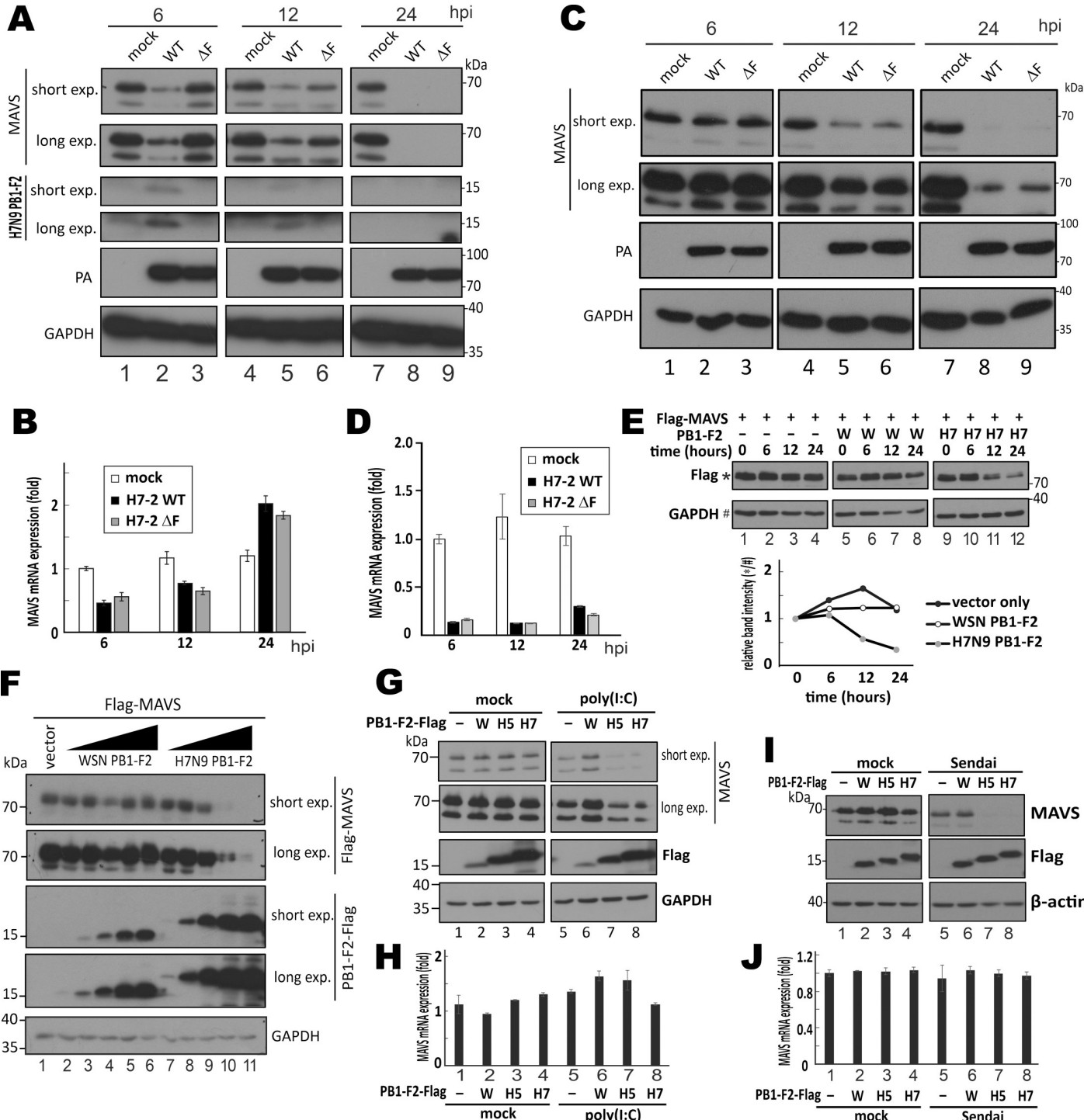

**Fig 5. Facilitation of MAVS degradation by H7N9 PB1-F2 in immune-stimulated cells.** (**A and B**) Impact of H7N9 PB1-F2 on MAVS protein level. THP-1 cells were infected with H7-2 WT and H7-2ΔF viruses at MOI = 1. At 6, 12, and 24 hpi, infected cells were harvested for total protein extraction, SDS-PAGE and Western blot analysis with anti-MAVS and anti-H7N9 PB1-F2 (A). PA and GAPDH served as normalization controls for viral infection and loading. Harvested cells were also subjected to RT-qPCR analysis of MAVS mRNA level relative to that of GAPDH mRNA using comparative Ct method (B). (**C and D**) Impact of WSN PB1-F2 on MAVS protein level. THP-1 cells were infected with WSN WT and WSN ΔF viruses at MOI = 1. MAVS protein and mRNA were analyzed as above. (**E**) HEK293T cells in 6-well plates were co-transfected with 1 μg pEF-Bos-Flag-MAVS and 0.2 μg PB1-F2 expression constructs. After 24 hours, 100 μg/mL cycloheximide (CHX) was added. Cells were harvested for total protein extraction at 0, 6, 12 and 24 hours after drug treatment for SDS-PAGE followed by Western blot analysis with anti-Flag. Relative band intensity of MAVS over GAPDH was plotted in the lower panel. (**F**) HEK293T cells were transfected with 0.5 μg pEF-Bos-Flag-MAVS together with increasing dosage of expression construct for PB1-F2-Flag of WSN and H7N9 (0, 0.25, 0.5, 1, 1.5 and 2 μg). After 48 hours, cells were harvested for total protein

extraction followed by SDS-PAGE and Western blot analysis with anti-Flag for detection of both MAVS (70 kDa) and PB1-F2 (15–20 kDa) normalized to GAPDH. (**G and H**) HEK293T cells in 6-well plates were transfected with 1.5 μg PB1-F2 expression constructs. After 24 hours, cells were either mock-transfected or transfected with 1 μg/mL poly (I:C) for 18 hours before harvested for total protein extraction followed by SDS-PAGE and Western blot analysis with anti-MAVS and anti-Flag (G) or for RT-qPCR analysis of MAVS mRNA (H). (**I and J**) HEK293T cells in 6-well plates were transfected with 1.5 μg PB1-F2 expression constructs. After 24 hours, cells were either mock-infected or infected with Sendai virus (100 hemagglutinating units/ml) for 18 hours before harvested for total protein extraction followed by SDS-PAGE and Western blot analysis with anti-MAVS and anti-Flag (I) or for RT-qPCR analysis of MAVS mRNA relative to HPRT mRNA by comparative Ct method (J). Bars represent means ± SD of triplicate experiments.

destabilized MAVS protein, cells in which MAVS and H7N9 PB1-F2 are overexpressed were treated with proteasome inhibitor MG132 and lysosome inhibitor bafilomycin A1. The administration of MG132 or bafilomycin A1 alone for 6 and 16 hours could not fully rescue the suppression of MAVS protein expression by H7N9 PB1-F2 (Fig 6A, lane 3 vs 1, and lane 6 vs 4). However, in the presence of both MG132 and bafilomycin A1 for 16 hours, the level of MAVS protein was gradually restored (Fig 6A, lane 9 vs 7). This result suggested that both proteasomal and lysosomal degradation might contribute to H7N9 PB1-F2-induced destabilization of MAVS. The destabilizing effect of WSN PB1-F2 on MAVS was not evident at the early time point of 6 hours and less prominent than that of H7N9 PB1-F2 when cells were treated for 16 hours, but the same trend in response to MG132, bafilomycin A1 and their combination was also seen (Fig 6A, lanes 2, 5 and 8).

To test if MAVS aggregation might be important for H7N9 PB1-F2-induced degradation, aggregation-defective mutants of MAVS [50, 53], including E26A, W56R and R64, 65A, were generated. Whereas H7N9 PB1-F2 was fully competent in destabilizing wild-type MAVS, its influence on all aggregation-defective mutants of MAVS was much less pronounced (Fig 6B, lanes 6, 9 and 12 vs 3). Collectively, the above results suggested that H7N9 PB1-F2 might target MAVS for proteasomal and lysosomal degradation in an aggregation-dependent manner. In other words, H7N9 PB1-F2 could specifically destabilize the aggregated and activated form of MAVS.

## Suppression of MAVS aggregation by H7N9 PB1-F2

To clarify exactly how H7N9 PB1-F2 might affect MAVS aggregation, we performed SDD-AGE analysis as previously described [35, 54]. Since expression of H7N9 PB1-F2 led to reduction of the total level of MAVS protein (Fig 5), HEK293T cells were overexpressed with MAVS and H7N9 PB1-F2 in the ratio of 1 μg to 0.2 μg so that steady-state level of overexpressed MAVS became stable (Fig 5F). WSN PB1-F2 and H5N1 PB1-F2 were included for comparison. H7N9 PB1-F2 showed the most prominent suppressive effect on MAVS aggregation, followed by H5N1 PB1-F2 and then WSN PB1-F2 (Fig 7A, lane 4 vs 2 and 3). To check for PB1-F2-MAVS interaction, co-immunoprecipitation was performed in cells in which MAVS was overexpressed from a stronger mammalian expression vector pCAGEN, which produced 7-fold more induction of IFN reporter activity than pEF-Bos (S4 Fig). Steady-state level of MAVS remained constant when cells were transfected with 1 μg pCAGEN-MAVS and 0.5 μg PB1-F2 plasmid (Fig 7B). Interestingly, H7N9 PB1-F2 did not interact with MAVS protein (Fig 7B, lane 8 vs 6). In contrast, MAVS was weakly detected in the H5N1 PB1-F2 precipitate and the association between WSN PB1-F2 and MAVS was much more pronounced (Fig 7B, lanes 6 and 7).

Confocal microscopy was next performed as previously described to visualize PB1-F2 and MAVS aggregation [55]. Aggregated MAVS presented as intense fluorescent foci or structures was seen to colocalize with fused mitochondria in HEK293T cells overexpressing MAVS (Fig 7C, panels 1–5). Consistent with results from co-immunoprecipitation (Fig 7B), non-aggregating WSN PB1-F2 was found to colocalize with MAVS aggregates and fused mitochondria (Fig

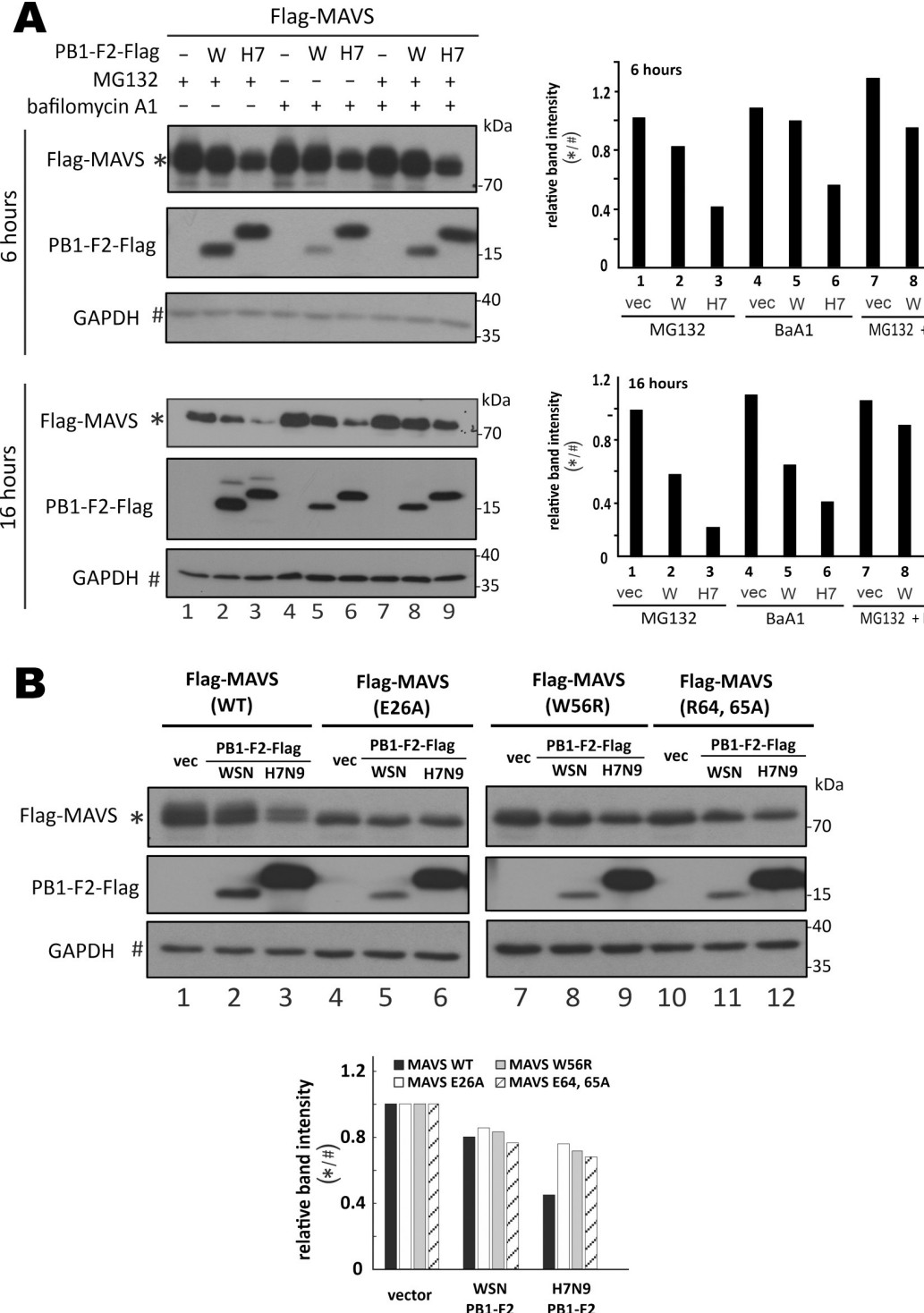

**Fig 6. H7N9 PB1-F2-induced destabilization of MAVS protein aggregate for proteasomal and lysosomal degradation.** (**A**) Treatment with proteasome and lysosome inhibitors. HEK293T cells in 6-well plates were co-transfected with 1 µg PB1-F2-Flag expression construct and 1 µg pEF-Bos-Flag-MAVS. After 24 hours, 20 µM MG132, 100 nM bafilomycin A1 (BaA1) or their combination was added for 6 or 16 hours before cells were harvested for total protein extraction and SDS-PAGE Western blot analysis against anti-Flag. Relative band intensity of MAVS over GAPDH was plotted in the right panel. (**B**) Analysis of aggregation-defective mutants of MAVS. HEK293T cells were co-transfected with 0.5 µg PB1-F2 expression construct and 0.5 µg pEF-Bos-Flag-MAVS WT, E26A, W56R or R64, 65A. After 48 hours, total protein was extracted from the cells and subjected to SDS-PAGE followed by Western blot analysis with anti-Flag for detection of MAVS at 70 kDa and PB1-F2 at 15-20kDa. Relative

band intensity of MAVS over GAPDH was plotted in the lower panel. Three independent experiments were performed with similar results.

7C, panels 6–10). However, minimal co-localization between aggregation-competent H7N9 PB1-F2 and MAVS was observed (Fig 7C, panels 11–15). Indeed, H7N9 PB1-F2 still localized to mitochondrial clusters as shown in Fig 2A, but much less MAVS aggregates were seen and

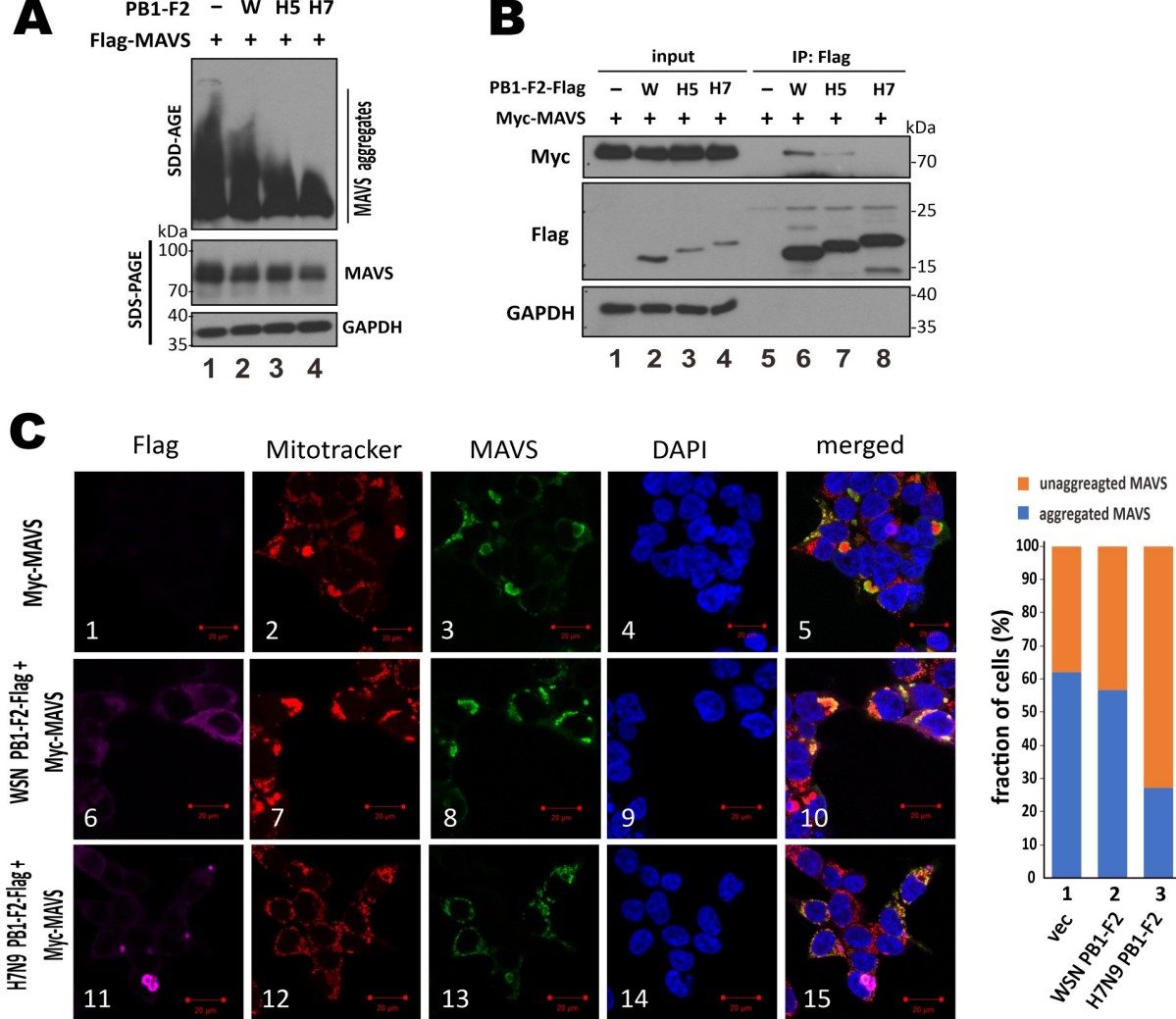

**Fig 7. Suppression of MAVS aggregation by H7N9 PB1-F2.** (**A**) SDD-AGE analysis of MAVS aggregation. HEK293T cells in 6-well plates were co-transfected with 1 μg pEF-Bos-Flag-MAVS and 0.2 μg expression construct for PB1-F2 of WSN, H5N1 or H7N9 virus. After 48 hours, cells were harvested for SDD-AGE or SDS-PAGE followed by Western blot analysis with anti-MAVS. (**B**) Co-immunoprecipitation assay for PB1-F2-MAVS association. HEK293T cells in 60mm dishes were co-transfected with 1 μg pCAGEN-myc-MAVS and 0.5 μg PB1-F2-Flag expression construct. After 48 hours, cells were harvested for co-immunoprecipitation with anti-Flag (IP: Flag). Both input and immunoprecipitates were analyzed by SDS-PAGE and Western blot analysis with anti-Myc for MAVS, anti-Flag for PB1-F2 and anti-GAPDH for normalization. (**C**) Distribution of unaggregated MAVS on fissioned mitochondria in cells expressing H7N9 PB1-F2. HEK293T cells in 6-well plates were co-transfected with 0.25 μg of CAGEN-V5-MAVS and 0.25 μg of PB1-F2-Flag expression constructs. After 48 hours, cells were stained with 500 nM Mitotracker Red CMXRos for 30 min and then fixed with 4% paraformaldehyde and probed with anti-Flag for PB1-F2, anti-MAVS and DAPI. The stained cells were analyzed by confocal microscopy. Fractions of cells with unaggregated and aggregated MAVS were calculated by counting 100 cells per sample. Aggregated MAVS or PB1-F2 was visually defined as concentrated dots or structures with intense MAVS or PB1-F2 signal. Bars, 20 μm. Similar results were obtained from three independent experiments.

MAVS was found to distribute more evenly on fissioned or fragmented mitochondria, which were rarely observed in the vector control and the WSN PB1-F2 groups (Fig 7C, panels 11–15 vs 1–5 and 6–10). Notably, H7N9 PB1-F2 aggregates were seen in the vicinity of MAVS staining, which substantially overlapped with that of Mitotracker (Fig 7C, panels 11–15). Quantitatively, by counting cells co-expressing MAVS and PB1-F2, about 60% of cells in the vector control and WSN PB1-F2 groups were found to harbor aggregated MAVS. This percentage dropped to around 25% in H7N9 PB1-F2-expressing cells (Fig 7C, right chart). That is to say, expression of H7N9 PB1-F2 effectively suppressed MAVS aggregation.

## Recruitment of TRAF3 but not TRAF6-TBK1-IKKε signalosome by MAVS in the presence of H7N9 PB1-F2

To derive further mechanistic insight on how H7N9 PB1-F2 suppresses MAVS, co-immuno-precipitation was performed to check for the interaction of MAVS with downstream effectors TRAF3, TRAF6, TBK1 and IKKε. TRAF3 was found to bind with multimerized MAVS through TRAF3IP3 independently of CARD domain-dependent aggregation [56, 57]. Instead, TRAF6-MAVS interaction requires complete MAVS aggregation through CARD-CARD interaction [51, 52]. MAVS interaction with TBK1 and IKKε is essential for formation of mature MAVS signalosome formation [58]. Half transfection dose of PB1-F2 was used in this experiment to achieve constant steady-state expression level of MAVS. Indeed, we found that H7N9 PB1-F2 did not affect the interaction of MAVS with TRAF3 (Fig 8A, lane 4 vs 2), but abolished MAVS interaction with TRAF6 (Fig 8B, lane 4 vs 2). Moreover, expression of H7N9 PB1-F2 potently suppressed MAVS-TBK1 (Fig 8C, lane 4 vs 2) and MAVS-IKKε interaction (Fig 8D, lane 4 vs 2). Compared to H7N9 PB1-F2, WSN PB1-F2 was much weaker in the suppression of MAVS-TRAF6, MAVS-TBK1 and MAVS-IKKε interaction (Fig 8A–8D). These results suggested that MAVS failed to recruit downstream effectors TRAF6, TBK1 and IKKε but was still capable of interacting with TRAF3 when H7N9 PB1-F2 was expressed.

## Influence of H7N9 PB1-F2 on TRIM31 binding and polyubiquitination of MAVS

Protein aggregation has an inhibitory effect on proteasomal degradation [59–61]. In addition, MAVS localized to fissioned mitochondria (Fig 7C) might also be prone to mitophagic or lysosomal degradation [62]. Both proteasomal and lysosomal degradation of MAVS is strictly controlled by polyubiquitination. Extensive K48- and K27-polyubiquitination of MAVS facilitates its proteasomal [63–68] and lysosomal [69, 70] degradation, respectively. K63-polyubiquitination, however, is important for MAVS aggregation and consequent activation [71]. To determine how H7N9 PB1-F2 triggers MAVS degradation, ubiquitination assay was performed. General ubiquitination of MAVS was only reduced slightly when H7N9 PB1-F2 was expressed (Fig 9A, lane 4 vs 2). By using ubiquitin K48, K63 and K27 mutants, it was found that K48- and K27-polyubiquitination of MAVS was largely unaffected by H7N9 PB1-F2 (Fig 9B, lanes 3 vs 2, and lane 9 vs 8). However, K63-polyubiqutination of MAVS was severely impaired in the presence of H7N9 PB1-F2 (Fig 9B, lane 6 vs 5). Unlike H7N9 PB1-F2, WSN PB1-F2, which minimally affected MAVS aggregation (Fig 7), did not mitigate K48-, K63- or K27 polyubiquitination of MAVS (Fig 9C, lane 3 vs 2, lane 7 vs 6, and lane 11 vs 10). Thus, H7N9 PB1-F2 prevented K63-polyubiquitination and aggregation of MAVS, but directed it for proteasomal and lysosomal degradation.

TRIM31 is the E3 ubiquitin ligase promoting K63-polyubiquitination, aggregation and activation of MAVS [71]. Aggregation-prone mitochondrial protein FAF1 suppresses RIG-I signaling by blocking TRIM31-MAVS interaction and in turn K63-polyubiquitination of MAVS

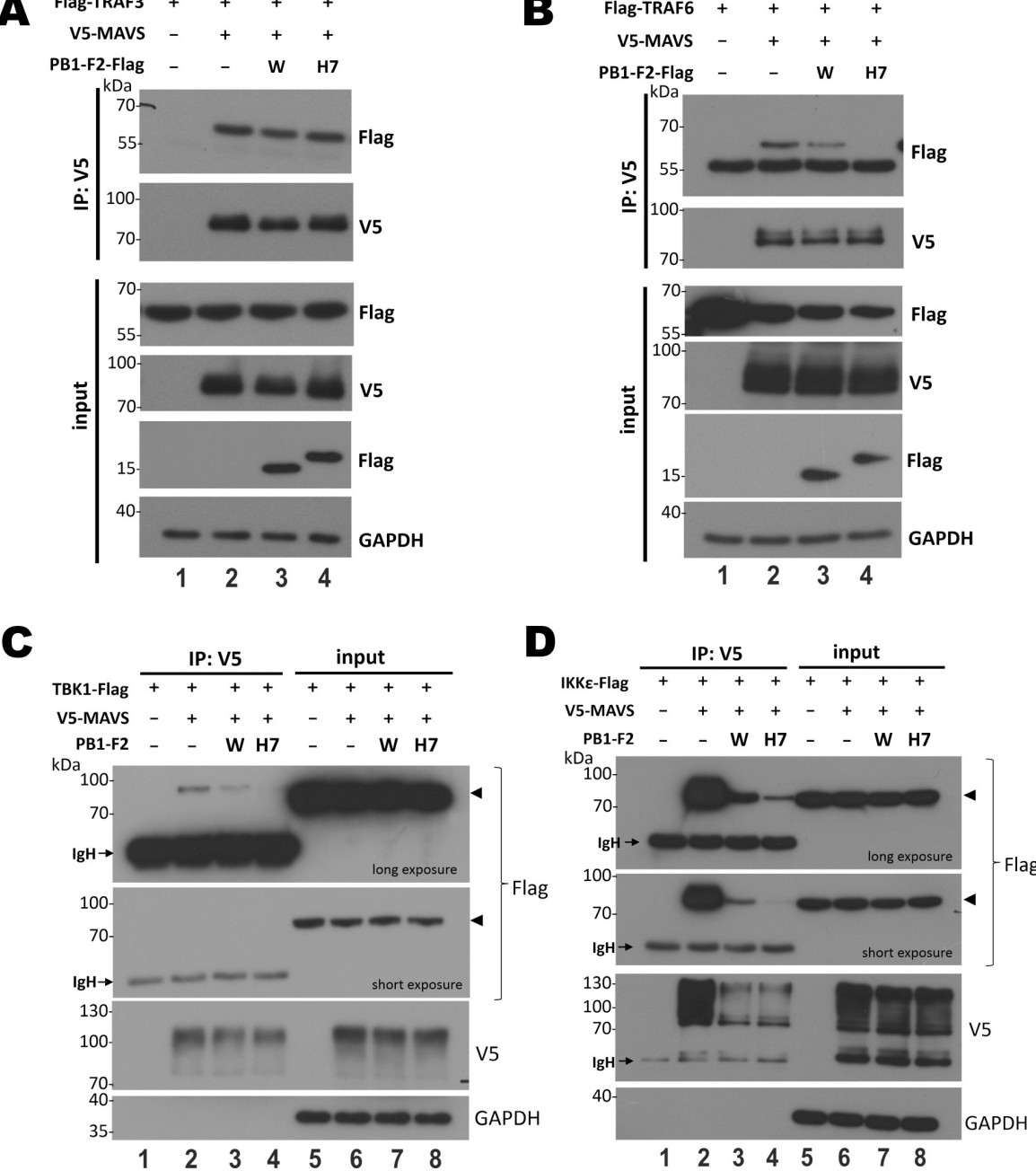

**Fig 8. Interaction of MAVS with TRAF3 but not TRAF6-TBK1-IKKε signalosome in the presence of H7N9 PB1-F2.** HEK293T cells in 60mm dishes were co-transfected with 0.5 μg pCAGEN-V5-MAVS, 0.25 μg PB1-F2 expression construct and 0.5 μg expression construct for Flag-TRAF3 (**A**), Flag-TRAF6 (**B**), Flag-TBK1 (**C**) or Flag-IKKε (**D**). After 48 hours, cells were harvested for co-immunoprecipitation with anti-V5 (IP: V5). Both input and immunoprecipitates were analyzed by SDS-PAGE and Western blot analysis with anti-V5 for MAVS, anti-Flag for TRAF3 (A), TRAF6 (B), TBK1 (C) and IKKε (D). GAPDH served as loading control. Arrowheads point to the target TBK1-Flag and IKKε-Flag bands. IgH: immunoglobulin heavy chain. Three independent experiments were carried out and similar results were obtained.

[43]. As aggregation-prone H7N9 PB1-F2 was found to suppress K63-polyubiquitination of MAVS (Fig 9B and 9C), we asked whether H7N9 PB1-F2 might also affect TRIM31-MAVS interaction. Indeed, by reciprocal co-immunoprecipitation, we found that H7N9 PB1-F2

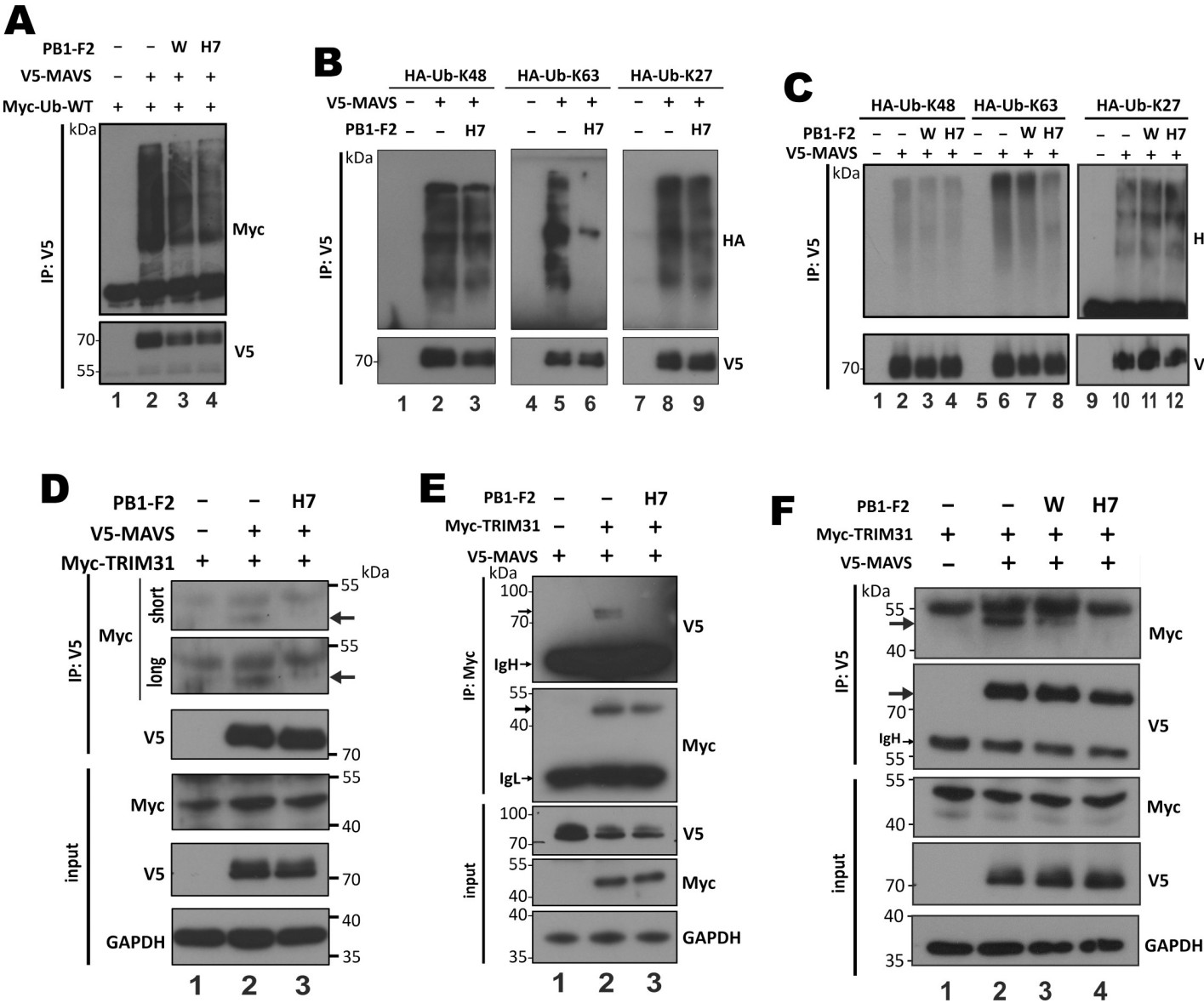

**Fig 9. Suppression of TRIM31-mediated K63 polyubiquitination of MAVS by H7N9 PB1-F2.** (**A**) HEK293T cells in 60mm dishes were transfected with 0.5 μg expression construct for Myc-Ub-WT, 0.5 μg pCAGEN-V5-MAVS and 0.25 μg PB1-F2 expression construct. After 48 hours, cells were harvested for immunoprecipitation with anti-V5 (IP: V5). Immunoprecipitates were analyzed with SDS-PAGE and Western blot analysis with anti-Myc for ubiquitin chain and anti-V5 for MAVS. (**B**) HEK293T cells in 60mm dishes were transfected with 2 μg expression construct for HA-Ub-mutants (K48, K63 or K27), 0.5 μg pCAGEN-myc-MAVS and 0.25 μg PB1-F2 expression construct. Immunoprecipitation and SDS-PAGE were performed as in (A). Western blot analysis was carried out with anti-HA for Ub mutant chains and anti-V5 for MAVS. (**C**) Same as in (B) except inclusion of WSN PB1-F2. (**D-F**) HEK293T cells in 60mm dishes were transfected with 2 μg expression construct for myc-TRIM31, 0.2 μg pCAGEN-myc-MAVS and 0.1 μg PB1-F2 expression construct. After 48 hours, cells were harvested for co-immunoprecipitation with anti-V5 (D and F) or anti-Myc (E) as indicated (IP: V5 or IP: Myc). Both input and immunoprecipitates were analyzed by SDS-PAGE and Western blot analysis with anti-V5 for MAVS, anti-Myc for TRIM31 and anti-GAPDH for normalization. Arrows point to the target Myc-TRIM31 and V5-MAVS bands. Two exposures (long and short) of the same blot are presented in (D). IgH: immunoglobulin heavy chain. IgL: immunoglobulin light chain. Results were representative of three independent experiments.

blocked TRIM31-MAVS interaction (Fig 9D and 9E). Compared to H7N9 PB1-F2, WSN PB1-F2 could minimally affect TRIM31-MAVS interaction (Fig 9F). Thus, like FAF1, H7N9 PB1-F2 perturbed TRIM31-MAVS interaction required for K63-polyubiquitination and aggregation of MAVS.

## Discussion

In this study, we defined a novel and virus subtype-specific mechanism by which PB1-F2 protein of the 2013 H7N9 virus prevents aggregation and activation of MAVS leading to suppression of innate immune response. H7N9 PB1-F2 had unique properties including higher aggregation potential (Fig 1), higher resistance to proteolysis (Fig 1) and tendency to accumulate on clustered mitochondria (Fig 2). H7N9 PB1-F2 potently inhibited K63-polyubiquitination (Fig 9), aggregation (Fig 7) as well as TRIM31-binding (Fig 9), TRAF6-engaging (Fig 8) and IFN-inducing (Fig 3) activities of MAVS. It also targeted MAVS to proteasomal and lysosomal degradation (Figs 5 and 6). Importantly, MAVS was stabilized and IFNβ was induced more robustly by H7N9 PB1-F2-defective recombinant virus (Figs 4 and 5). Our findings highlight the importance of PB1-F2 in pathogenicity of avian influenza A viruses H7N9 and H5N1.

We demonstrated the aggregation-prone property of H7N9 PB1-F2 (Fig 1). Since aggregated proteins are known to be less susceptible to proteolysis [59–61], the higher aggregation potential of H7N9 PB1-F2 is generally consistent with its higher resistance to proteolysis and increased protein stability (Fig 1). We also showed that H7N9 PB1-F2 primarily affects aggregation of MAVS, which was consistently supported by several lines of evidence. First, MAVS aggregation was severely compromised when H7N9 PB1-F2 was expressed (Fig 7A). Second, MAVS aggregates on mitochondria seen in confocal images disappeared when H7N9 PB1-F2 was expressed (Fig 7C). Third, unaggregated MAVS in the presence of H7N9 PB1-F2 is more prone to proteasomal and lysosomal degradation in transfected and infected cells (Figs 5 and 6A). This is also generally consistent with the notion that aggregated proteins are less susceptible to proteolysis [59–61]. Fourth, H7N9 PB1-F2 had little suppressive effect on aggregation-defective mutants of MAVS (Fig 6B). Fifth, MAVS retained its ability to interact with TRAF3 but lost its ability to interact with TRAF6, TBK1 and IKKε in the presence of H7N9 PB1-F2 (Fig 8). Whereas the interaction with TRAF3 requires multimerization but not aggregation of MAVS, MAVS aggregation is a prerequisite for the interaction with TRAF6, TBK1 and IKKε [51, 52, 58]. Finally, H7N9 PB1-F2 suppressed TRIM31 recruitment to MAVS and K63-polyubiquitination of MAVS (Fig 9), both of which are required for aggregation and activation of MAVS. Whether and how the aggregation of H7N9 PB1-F2 might control its suppression of MAVS aggregation and activation merit further investigations.

MAVS was originally proposed to undergo spontaneous prion-like aggregation upon activation [51]. However, it is now known that TRIM31-mediated K63-polyubiquitination of MAVS is essential for aggregation [71]. The TRIM31-MAVS axis is stringently regulated by multiple host factors such as aggregation-prone mitochondrial adaptor protein FAF1 [43], K63-deubiquitinase YOD1 [72], RNF34 that promotes K63-K27 chain transfer [68], O-GlcNAc transferase that catalyzes O-GlcNAcylation [73, 74], SIRT5 that catalyzes lysine desuccinylation [75], Rac-1 that affects protein geranylgeranylation [76] and SNX8 that controls cholesterol levels [77]. In addition, MAVS aggregation can be suppressed by lactate [78] and TRIM31 is known to bind with phosphotidyethanolamine [79]. Here we showed that viral protein PB1-F2 of avian influenza virus H7N9 formed protein aggregates on mitochondria, perturbed TRIM31-MAVS interaction and directed unaggregated MAVS for proteasomal and lysosomal degradation. H7N9 PB1-F2 shares some functional similarities with cellular protein FAF1 [43]. Both localize primarily to mitochondria and are aggregation-prone. However, FAF1 does not enhance MAVS degradation. Since FAF1 interacts tightly with MAVS, they likely co-aggregate to form a stable FAF1-MAVS complex. In contrast, no interaction between H7N9 PB1-F2 and MAVS was detected (Fig 7B). Instead, H7N9 PB1-F2 aggregated and localized to distinct sub-compartment of clustered mitochondria (Fig 2A), adjacent to the sub-

compartment harboring unaggregated MAVS on fissioned or fragmented mitochondria (Fig 7C), where MAVS was prone to degradation.

It is still an open question as to how H7N9 PB1-F2 abolishes MAVS-TRIM31 interaction. TRIM31 is not a mitochondrial membrane protein but attached to mitochondria through an unknown factor [79]. Sendai virus infection enhances MAVS-TRIM31 complex formation [71], suggesting that MAVS aggregation favors TRIM31 mitochondrial localization and MAVS-TRIM31 interaction. However, H7N9 PB1-F2 formed amyloid aggregates on mitochondria leading to alteration of mitochondrial morphology (Fig 2A), which might pose steric hindrance to aggregation of MAVS, which did not interact with H7N9 PB1-F2 directly (Fig 7B). Plausibly, MAVS aggregation was hindered but its multimeric structure remained intact, enabling it to interact with TRAF3 only (Fig 8A) but not TRAF6, TBK1, IKKɛ or TRIM31 (Figs 8B–8D, 9D and 9E). Like a negative feedback loop, weakening of TRIM31-MAVS interaction by H7N9 PB1-F2 suppressed K63-polyubiquitintion, aggregation and activation of MAVS (Fig 9B and 9C). It will be of great interest to determine whether H7N9 PB1-F2 might directly target TRIM31 or other essential components that mediate MAVS-TRIM31 complex formation, including YOD1, SIRT5 and SNX8. Particularly, whether and how H7N9 PB1-F2 might affect phospholipids in mitochondrial membrane merit further investigations. Since the ubiquitination of H7N9 PB1-F2 is known to be required for its biological function [38], the role of TRIM31 in this process might also be worthy of experimental validation.

The suppressive effect of PB1-F2 on MAVS aggregation and stability is virus subtype-specific. H7N9 PB1-F2 but not WSN PB1-F2 was capable of destabilizing MAVS and preventing its aggregation. Although the activity profiles of H7N9 PB1-F2 and H5N1 PB1-F2 were distinct, both proteins were competent to form aggregates on mitochondria (Figs 1 and 2) and to destabilize MAVS (Fig 5G and 5I). Thus, the two pathogenic avian influenza viruses share a similar MAVS-destabilizing property. It will be of interest to determine the residues and regions in PB1-F2 proteins of H7N9 and H5N1 viruses required for aggregation. As predicted by PASTA 2.0 (Fig 1C and S1 Fig), however, anti-parallel pairing that promotes protein aggregation spans over a wide C-terminal region of both proteins, probably involving multiple residues. Structural analysis of PB1-F2 is required to clarify their exact roles. Other subtype-specific properties of PB1-F2 have been well documented [11, 12, 80]. Although originally thought to be properties of all PB1-F2 proteins [18–20], abilities to sequester MAVS and to decrease ΔΨm are obligatory to PR8 or WSN virus (Figs 2 and 7). PB1-F2 of the 1918 pandemic strain is known to co-degrade with DDX3X, leading to inhibition of TBK1 [21]. Whereas PB1-F2 of the 1918 strain is a fast-turnover protein similar to WSN PB1-F2, H7N9 PB1-F2 is much stabilized (Fig 1). Whether PB1-F2 of the 1918 strain might also be aggregation-prone and whether H7N9 PB1-F2 could also target DDX3X for degradation merit further comparison and analysis. Under a strong selection pressure [12], PB1-F2 proteins of different subtypes might have been evolved to target MAVS and other adaptor proteins in the RIG-I signaling pathway using different strategies. Instead of binding with and sequestering MAVS, H7N9 PB1-F2 suppressed its aggregation and destabilized it. Notably, the suppressive effect of H7N9 PB1-F2 was only seen when MAVS was activated (Fig 5). In other words, H7N9 PB1-F2 had no influence on basal activity of MAVS in the absence of immune stimulation. Plausibly, H7N9 PB1-F2 functions to prevent excessive activation of MAVS signaling.

PB1-F2 amyloid formation occurs as early as at 8 hpi as demonstrated with WSN virus at an MOI of 0.1 [26]. As it had higher aggregation propensity than WSN PB1-F2 (Fig 1), H7N9 PB1-F2 might aggregate more rapidly. In our setting, H7-2 WT infection at an MOI of 1 strikingly reduced MAVS protein level as early as at 6 hpi (Fig 5A), leading to an inhibition of type I IFN production at 12 and 24 hpi (Fig 4B). H7N9 PB1-F2 was abundantly expressed at 6 hpi but started to decline from 12 to 24 hpi (Fig 5A). The expression of H7N9 PB1-F2 likely

plateaued at around 6 hpi, before which H7N9 PB1-F2 was already aggregating on mitochondria to dampen MAVS aggregation, thereby blocking propagation of antiviral signal. Electrochemical study as described [25] can be performed to investigate further as to how H7N9 PB1-F2 aggregates in infected cells.

Lethal H5N1 and H7N9 infection arises from cytokine storm [1, 5]. Here we found that H7N9 PB1-F2 potently suppressed type I IFN expression but might not affect production of proinflammatory cytokines such as TNFα (Fig 4B and 4C). This suggests that suppression of MAVS-mediated antiviral signaling by H7N9 PB1-F2 might exert a specific effect on type I IFN production but minimally affect the overall proinflammatory response. This is compatible with the pattern of IFN and cytokine expression during pathologic cytokine storm of H7N9 infection, with profound proinflammatory cytokine production but attenuated type I IFN production and signaling [81, 82]. This is also in keeping with our recent finding that WSN PB1-F2 is capable of suppressing inflammasome activation induced by various stimuli, whereas H7N9 PB1-F2 exerts a more restricted inhibitory effect on RNA-induced interleukin 1β maturation and secretion [83].

Recombinant influenza A viruses deficient of another IFN antagonist NS1 have been evaluated for use as the backbone for construction of candidate attenuated vaccines due to its potent type I IFN-inducing property [84]. However, severely attenuated replication of the NS1-deleted viruses poses a challenge to the preparation of seed virus [85]. A DelNS1 system with adaptive mutations to support virus replication in eggs and MDCK cells has recently been reported for making optimal live attenuated influenza vaccines [86]. Although the elevation of the type I IFN-inducing activity of the recombinant H7N9 PB1-F2-deficient virus was mild (Fig 4B), its replication was not affected in THP-1 cells (Fig 4B) and modestly attenuated in A549 cells (Fig 4C). It will be of great interest to test whether PB1-F2 knockout could serve as another strategy for construction of attenuated vaccines. Particularly, whether double knockout of NS1 and PB1-F2 might have any advantages in boosting innate immunity warrants experimental validation.

## Materials and methods

### Plasmids

For PB1-F2 expression constructs, PB1-F2-coding sequences of A/PR/8/1934 (PR8), A/WSN/1933 TS61 (WSN), A/Vietnam/1203/2004 (H5N1) and A/Zhejiang/DTID-ZJU01/2013 (H7N9) were subcloned into pCAGEN vector to express PB1-F2 with three Flag tags at the C terminus (PB1-F2-Flag). pEF-Bos-Flag-MAVS was generated as previously described [43]. Aggregation-defective mutants of MAVS (E26A, W56R and R64, 65A) were generated on pEF-Bos-Flag-MAVS through site-directed mutagenesis using Q5$^{®}$ Site-Directed Mutagenesis Kit (New England Biolabs). MAVS-coding sequence was subcloned into pCAGEN to generate pCAGEN-Myc-MAVS and pCAGEN-V5-MAVS. Expression constructs for RIG-IN, TRAF3, TRAF6, TBK1, IKKε, IRF3-5D and MyD88 as well as reporter constructs pIFNβ-Luc, pISRE--Luc, pIRF3-Luc, pκB-Luc and pRL-Tk were as described [45]. TRIM31-coding sequence was subcloned from I.M.A.G.E. clone 3846513 into pCMV-Tag3B vector. Eight-plasmid system for recombinant influenza A virus based on pHW2000 vector have been described [87]. H7-2 WT and H7-2ΔF viruses were made with PB2, PB1, PA, NP, M and NS from H7N9 plus HA and NA from WSN (Fig 5A). H7N9 PB1-F2 knockout (Fig 5A) was achieved by using Q5$^{®}$ Site-Directed Mutagenesis Kit. Mammalian expression constructs for ubiquitin and its mutants have been described [88]. The 12-plasmid reverse genetic system, including 8 pHH21 constructs for vRNA segments and 4 protein expression constructs driven by pCAGGS for PB2, PB1, PA, and NP, for production of WSN WT (A/WSN/1933 TS61) and WSN ΔF viruses,

was a gift from Dr. Yoshihiro Kawaoka (University of Wisconsin, Madison, WI, USA) [89]. PB1-F2 knockout was achieved by using Q5® Site-Directed Mutagenesis Kit.

## Primary antibodies

Commercial primary antibodies used in this study included mouse anti-Flag (M2; Sigma-Aldrich), mouse anti-V5 (Invitrogen), mouse anti-Myc (9E10; Santa Cruz), mouse anti-HA (F7; Santa Cruz), mouse anti-GAPDH (6C5; Santa Cruz), mouse anti-β-actin (Sigma-Aldrich), mouse anti-α-tubulin (Sigma-Aldrich), mouse anti-MAVS (E3; Santa Cruz), rabbit anti-MAVS (D5A9E; Cell Signaling), rabbit anti-COXIV (3E11; Cell Signaling) and rabbit anti-PA (PA5-32224; Invitrogen). Mouse monoclonal antibody to PB1-F2 was raised using the custom antibody production service of Genscript (Piscataway, New Jersey) with full-length recombinant protein of PB1-F2 of A/Zhejiang/DTID-ZJU01/2013 (H7N9) expressed from *E. coli* as immunogen.

## Chemicals and drugs

MG132 was purchased from Abcam. Bafilomycin A1 was from Invivogen. CCCP, poly (I:C) and cycloheximide were from Sigma. 4',6-diamidino-2-phenylindole (DAPI), Mitotracker Red CMV Rox and JC-1 dye were from ThermoFisher.

## Cell culture

Cell lines HEK293T, A549, THP-1, DF-1 and MDCK cell lines were from ATCC. HEK293T, A549 and DF-1 cells were maintained in Dulbecco's Modified Eagle's Medium with 10% fetal bovine serum. MDCK cells were maintained in Minimum Essential Medium with 10% fetal bovine serum. THP-1 cells were maintained in RPMI 1640 with 10% fetal bovine serum, 1 mM sodium pyruvate and 1× non-essential amino acid. All cells were kept at 37°C in humidified incubators with 5% $CO_2$. Plasmid transfection was performed using GeneJuice (Novagen, USA) following manufacturer's protocol. For poly (I:C) transfection, Lipofectamine 2000 (ThermoFisher Scientific) was used as per manufacturer's protocol.

## Viruses

Sendai virus (VR-907) was purchased from ATCC. For recombinant influenza A viruses H7-2 WT and H7-2 ΔF, 8-plasmid reverse genetic rescue protocol was used. Briefly, HEK293T cells in 6-well plates with OPTI-MEM (ThermoFisher Scientific) were transfected with pHW2000 (500 ng) expressing PB2, PB1, PA, NP, M and NS genomic segments of H7N9 and HA and NA segments of WSN [90]. PB1 of H7-2 ΔF was created as shown in Fig 4A to disrupt PB1-F2 expression. After 24 hours, media of the transfected cells were changed to MEM with 1 μg /mL TPCK-treated trypsin. After another 48 hours, the conditioned media of HEK293T cells was diluted by serum-free MEM with 1 μg/mL TPCK-treated trypsin and incubated with confluent MDCK cells. After 48 hours, MDCK media containing recombinant viruses were collected, cleared by centrifugation at 3,000×g for 10 min and purified by centrifugal filter (100 kDa; Millipore) with MEM. The virus-containing media were either frozen at -80°C or used immediately for further analysis. For production of WSN WT and WSN ΔF, the 12-plasmid reverse genetic rescue protocol was used as previously described [91]. Briefly, 200 μl of seed virus-containing medium from HEK293T cells transfected with 12 plasmids for 72 hours was injected into 10-day-old embryonic eggs for virus propagation at 37°C humidified incubator. After 48 hours, allantoic fluid was collected from inoculated eggs. Both WSN WT and WSN ΔF were then further propagated twice in Vero-E6 cells to achieve sufficient viral titer for infection.

## General protein, RNA and reporter assays

Total protein sample was extracted as follows. Cells were lysed with RIPA lysis buffer (50 mM Tris-Cl pH 7.4, 100 mM NaCl, 0.1% SDS, 0.5% sodium deoxycholate, 2 mM EDTA and 1× EDTA-free-protease inhibitor) for 30 min on ice. After addition of 5× protein sample buffer (0.25 M Tris-Cl pH 6.8, 20% glycerol, 10% SDS and 1% saturated bromophenol blue), the lysate was boiled for 10 min. Insoluble debris was removed by centrifugation at 21,500 g for 10 min. SDS-PAGE and Western blot analysis were then performed as previously described [45].

Co-immunoprecipitation assay was performed as described [45]. For immunoprecipitation assay of detecting ubiquitin chains, RIPA lysis buffer was used to lyse target cells. After capturing target proteins onto antibody-beads complex, high salt wash buffer (50 mM Tris-Cl pH 7.4, 500 mM NaCl, 0.1% SDS, 0.5% sodium deoxycholate and 2 mM EDTA) was used to remove unwanted proteins bound. Proteins were eluted from beads with 2× protein sample buffer at 98°C for 10 min.

Luciferase reporter samples were extracted with passive lysis buffer provided by Promega. Dual luciferase reporter assay was performed as previously described [45].

Total RNA samples were either extracted with RNAiso PLUS (Takara) or RNeasy Mini Kit (Qiagen) following manufacturer's protocol. RT-qPCR was performed as previously described [45]. For reverse transcription of viral RNA, Uni12 primer with sequence 5′-AGCAAAAGCA GG-3′ was used. qPCR primers used in the study are listed below: MAVS: 5′-GTCACTTCCT GCTGAGA-3′ and 5′-TGCTCTGAATTCTCTCCT-3′. IFNβ: 5′-AGGACAGGATGAACTT TGAC-3′ and 5′-TGATAGACATTAGCCAGGA G-3′. TNFα: 5′-TCTGGGCAGGTCTACT TTGG-3′ and 5′-CCTAAGCCCCCAATTCTCTT-3′. vRNA S4: 5′-CGAGAAGAATGTGG CAGTGA-3′ and 5′-CTCCGTTGAATGTGTGGTTG-3′. vRNA S6: 5′-CAACCAAGGCAGC ATTACCT-3′ and 5′-ACGGATGGGACAAAGAGATG-3′. GAPDH: 5′-GGATTTGGTCG-TATTGGG-3′ and 5′-GGAAGATTGGTGATGGGATT-3′.

## SDD-AGE

SDD-AGE was performed as previously described [35, 54]. Briefly, HEK293T cells were lysed with cold SDD-AGE lysis buffer (20 mM Tris-HCl pH 8.0, 137 mM NaCl, 10% glycerol, 1% NP-40, 2 mM EDTA and 1× protease inhibitor) for 30 min. Appropriate amount of 5× SDD-AGE sample buffer (2.5× TBE, 2.5% SDS, 25% glycerol and 0.25% bromophenol blue) was then added to SDD-AGE lysate to reach 1×. The crude sample was passed through 25G needle 10 times at 4°C to break viscosity. The SDD-AGE protein sample was then separated by 2% agarose gel and transferred onto Immune-Blot® PVDF Membrane (BioRad). The membrane was blocked with 5% nonfat dried milk and then incubated with primary antibody. The blot was probed with secondary antibody conjugated with horseradish peroxidase. Protein aggregate ladder was detected with WesternBright ECL (Advansta).

## Protein solubility assay

Cells were lysed with cold RIPA lysis buffer for 30 min. A small portion of crude lysate was isolated out for total protein sample extraction as described. The crude lysate was then subjected for centrifugation at 21,500×g for 10 min at 4°C. RIPA supernatant was denatured in protein sample buffer in boiling water for 10 min. RIPA pellet was washed once with RIPA lysis buffer. 1× protein sample buffer was then added to RIPA pellet and boiled for 10 min. Undissolved debris was removed by further centrifugation at 21,500×g for 10 min. Protein samples of total protein, RIPA-soluble and RIPA pellet fractions were analyzed by SDS-PAGE and Western blotting.

## Immunofluorescent staining and confocal analysis

Cells on coverslips were either fixed with 4% paraformaldehyde for 15 min at room temperature followed by permeabilization with 0.2% Triton X100 for 5 min or with 1:1 methanol:acetone solution for 20 min at 20˚C. The cells were then washed three times with PBS and blocked with 3% BSA. After that, the cells were probed with primary antibody overnight at 4˚C. On the next day, cells were washed three times with PBS followed by probing with 0.5 μg/mL DAPI and secondary antibodies with appropriate origin and fluorophore. After three washes with PBS, the cells on coverslips were mounted and visualized with Carl Zeiss LSM 710 Confocal Microscope (Zeiss, Germany).

## JC-1 staining

Upon harvest, cells were trypsinized, washed once with PBS and resuspended in DMEM to reach $5 \times 10^6$ cells/ml. Cells were stained with 5 μg/ml JC-1 dye for 20 min. For CCCP control, cells were co-treated with JC-1 dye and 50 μM CCCP for 20 min. The cell suspensions were then washed with PBS and analyzed through fluorescence plate reader. Excitation at 550 nm and emission at 600 nm showed red signal representing JC-1 aggregate on healthy mitochondria. Excitation at 485 nm and emission at 535 nm showed green signal representing JC-1 monomer on depolarized mitochondria.

## Supporting information

**S1 Fig. Predicted parallel and anti-parallel cross β-pairing in WSN, H5N1 and H7N9 PB1-F2.** Amino acid sequences of WSN, H5N1 and H7N9 PB1-F2 were analyzed for putative cross β sheet structure using PASTA 2.0 (http://protein.bio.unipd.it/pasta2/). Predicted secondary structure and putative parallel or anti-parallel cross β-pairing were aligned to residues. P: parallel aggregation. A: anti-parallel aggregation. −: non-aggregating residue. D: disordered residue. O: ordered residue. H: α-helix residue. B: β-strand residue. C: coiled residue. (TIF)

**S2 Fig. Z-stack analysis for H7N9 PB1-F2 on mitochondrial aggregate.** HEK293T cells were transfected with 1 μg of H7N9 PB1-F2-Flag expression construct for 48 hours followed by Mitotracker Red CMXRos staining, fixation and immunostaining with anti-Flag and DAPI as in Fig 3A. The stained cells were analyzed by confocal microscopy for Z-stack imaging. Images 1 to 9 represented nine images captured through Z-axis perpendicular to the image plane spaced by 0.7 μm. Arrows indicated distinct mitochondrial clusters co-localized with H7N9 PB1-F2. (TIF)

**S3 Fig. Protein expression levels of PB1-F2s and the other proteins of interest in the luciferase assay.** Samples of the same conditioned HEK293T cells as described in Fig 3 were subjected to total protein sample extraction followed by SDS-PAGE and Western blot analysis. (**A**) Protein samples of HEK293T cells transfected with PB1-F2-Flag plasmids and infected with Sendai virus as in Fig 3A were tested for expression level of PB1-F2-Flag normalized to GAPDH level. (**B**) Protein samples of HEK293T cells transfected with PB1-F2-Flag plasmid and poly (I:C) as in Fig 3B were tested as above. (**C**) HEK293T cells were transfected with PB1-F2-Flag and GST-RIG-IN plasmids as in Fig 3C. (**D**) Cells were transfected with MyD88--His plasmid as in Fig 3D. (**E**) Flag-MAVS was expressed as in Fig 3E, 3H and 3I. (**F**) Flag-TBK1 was expressed as in Fig 3F. (**G**) IRF3-5D-V5 was expressed as in Fig 3G. (**H**) Cells were treated with IFNβ or transfected with Flag-MAVS plasmid as in Fig 3J. (**I**) H7N9 PB1-F2 66N-Flag or H7N9 PB1-F2-66S-Flag was expressed. Cells were infected with Sendai virus as in

Fig 3K. (**J**) Flag-MAVS was expressed as in Fig 3L.
(TIF)

**S4 Fig. pCAGEN-MAVS had stronger expression level than pEF-Bos-MAVS.** HEK293T
cells in 24-well plates were transfected with 50 ng pCAGEN-myc-MAVS or pEF-Bos-Flag-
MAVS plus 100 ng p125-Luc and 10 ng pRL-TK. After 48 hours, cells were harvested for dual-
luciferase reporter assay.
(TIF)

## Author Contributions

**Conceptualization:** Pak-Hin Hinson Cheung, Tak-Wang Terence Lee, Honglin Chen, Kwok-
Yung Yuen, Chi-Ping Chan, Dong-Yan Jin.

**Data curation:** Pak-Hin Hinson Cheung, Tak-Wang Terence Lee.

**Formal analysis:** Pak-Hin Hinson Cheung, Tak-Wang Terence Lee, Chun Kew, Honglin
Chen, Kwok-Yung Yuen, Chi-Ping Chan, Dong-Yan Jin.

**Funding acquisition:** Pak-Hin Hinson Cheung, Dong-Yan Jin.

**Investigation:** Pak-Hin Hinson Cheung, Tak-Wang Terence Lee.

**Methodology:** Pak-Hin Hinson Cheung, Tak-Wang Terence Lee, Chun Kew.

**Project administration:** Pak-Hin Hinson Cheung, Chi-Ping Chan, Dong-Yan Jin.

**Resources:** Chun Kew.

**Supervision:** Honglin Chen, Kwok-Yung Yuen, Chi-Ping Chan, Dong-Yan Jin.

**Visualization:** Pak-Hin Hinson Cheung, Tak-Wang Terence Lee.

**Writing – original draft:** Pak-Hin Hinson Cheung, Dong-Yan Jin.

**Writing – review & editing:** Pak-Hin Hinson Cheung, Tak-Wang Terence Lee, Chun Kew,
Honglin Chen, Kwok-Yung Yuen, Chi-Ping Chan, Dong-Yan Jin.

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
