## [Decision Letter · Decision Letter 0]

10 Mar 2020

Dear Professor Jin,

Thank you very much for submitting your manuscript "Virus subtype-specific suppression of MAVS aggregation and activation by PB1-F2 protein of influenza A (H7N9) virus" for consideration at PLOS Pathogens. As with all papers reviewed by the journal, your manuscript was reviewed by members of the editorial board and by several independent reviewers. In light of the reviews (below this email), we would like to invite the resubmission of a significantly-revised version that takes into account the reviewers' comments.

Please address all comments by the two reviewers.

We cannot make any decision about publication until we have seen the revised manuscript and your response to the reviewers' comments. Your revised manuscript is also likely to be sent to reviewers for further evaluation.

Sincerely,

Peter Palese

Associate Editor

PLOS Pathogens

Ron Fouchier

Section Editor

PLOS Pathogens

Kasturi Haldar

Editor-in-Chief

PLOS Pathogens

orcid.org/0000-0001-5065-158X

Michael Malim

Editor-in-Chief

PLOS Pathogens

orcid.org/0000-0002-7699-2064

Please address all comments by the two reviewers.

Reviewer's Responses to Questions

**Part I - Summary**

Reviewer #1: The accessory protein PB1-F2 of influenza A virus was previously shown to interfere with type I interferon induction at the level of MAVS through direct interaction with this central signaling hub. The authors provide experimental evidence to show a strain specific inhibition of MAVS function in H5 and H7 strain derived PB1-F2s. Mechanistically, the authors propose that especially H7 PB1-F2s support degradation of MAVS and disturb additionally the interaction with important downstream signaling molecules such as TRAF6. The hypothesis that PB1-F2 elicits a more potent phenotype in highly pathogenic influenza viruses is generally tempting and from a mechanistic point of view novel. It would fall in line with previous observations that pandemic viruses and highly pathogenic zoonotic viruses require PB1-F2 as a pathogenicity factor.

Reviewer #2: Submitted MS focuses on the fine molecular mechanism by which PB1-F2 (with special focus on avian H5/H7) modulates type I interferon response. PB1-F2 is widely accepted as a molecular pathogenicity determinant, with multifunctional properties in strain and host cell-dependent manner. Its mitochondrial localization 2003 by Gibbs (10.1128/JVI.77.13.7214-7224.2003) pointed attention towards innate immune response as mitochondria are well known to be involved. Early evidence (Le Gofic 2010) showed that PB1-F2 surpasses IFN in RIG-i dependent manner. Molecular mechanism remained elusive till ~2010 when Varga et al described mitochondrial antiviral signaling protein MAVS as RIG-I downstream effector of PB1F2 IFN suppression. As with any other PB1-f2 function, the extend of the IFN inhibition seems to be strain specific. The authors observed that Avian PB1F2 is a more potent IFN suppressor.

The MS confirmed 2011-2012 discoveries related to MAVS.

Importantly, several original observations in this MS deepen our understanding at the molecular level.

1. H7- and H5-PB1F2 but not H1- PB1F2 decreases aggregation and

2. destabilize MAVS at protein level via proteasomal and lysosomal degradation.

3. They also showed that this effect is more pronounced on aggregated MAVS (IFN response triggering form) by selectively hinder the ubiquitination of MAVS at K63. This residue ubiquitination promotes aggregation of MAVS.

4. H7N9 PB1F2 hinders interaction of the MAVS with TRIM31, ubiquitin ligase responsible for MAVS K63 ubiquitination. The authors correlated the level of PB1F2 amyloid fiber formation for H7 H5 H1 PB1f2 with its ability to destabilize MAVS.

The experimental design is of high quality, with appropriate controls (for vast majority experiments) and the results support claims and conclusions.

One drawback represents an inability to demonstrate that H7 PB1F2 to interact with MAVS, either I am missing something or it does not make much sense, authors need to clarify how is PB1F2 effect mediated?

In addition to plasmid co-transfection experiments, authors demonstrated MVAS mediated IFN inhibition in virus expressed PB1F2 context as well (that is super-significant as most works only focus on somewhat artificial co-transfection models as it is easier to control PB1-F2 expression).

It would be of great importance if authors could reproduce as many experiments as possible in the virus context. While PB1F2 with a taq expressed from the plasmid is of great convenience, antibodies are available

for Pb1-F2 (doi:10.1007/s00705-009-0479-5.).

This MS is of substantial readership interest as it revealed at a very fine molecular level how PB1F2 modulates IFN response and that avian Pb1F2 is a more potent IFN inhibitor.

**Part II – Major Issues: Key Experiments Required for Acceptance**

Reviewer #1: The study is largely based on over-expression experiments in 293T cells and a clear correlation between the protein amount of the three PB1-F2s and the respective phenotype investigated is obvious (e.g. Fig 6C and g, 7A/B). Interestingly, somewhat similar expression was achieved in Fig. 8C.

Luciferase reporter assays require transfection controls to assure comparable protein levels (e.g. Fig. 4a-l)

In several panels only H7 PB1-F2 is tested and the matching WSN PB1-F2 controls are missing. This needs to be included to point out the strain specific effects the authors are claiming (Fig 5, Fig. 10b d and e)

The quantification of Western blots in Fig. 7a does not appear to match with the actual bands.

All experiments require indication of independent repeats, biological and technical replicates and statistical methods applied in the respective figure legends.

The quality of western blots needs to be improved (some blots are clearly to faint (Fig 6g aMAVS, 7a GAPDH and numerous are plain overexposed 7a flag, 6a GAPDH, 6c flag, 9BC and D V5) . Often it is unclear if the signal is within the linear range. Additionally it remains elusive if the absolute amounts of PB1-F2 and MAVS are comparable to infection/cellular expression conditions.

How was aggregation of MAVS distinguished from overexpression in Fig 8C?

Reviewer #2: For the IF confocal microscopy Z stacking is essential to address the distribution of the PB1F2 at mitochondria/mitochondrial aggregates, the same slides could be used to rescan over several vertical planes to allow relevant co-localization distribution

line 287-289: how do authors explain that H7 PB1-f2 does not precipitate MAVS and has the most pronounced effect on it?

This Must be addressed, is there some third independent factor which promotes aggregation and H7 PB1-F2 sequestrate it??? Or...

line 314-317

If H7Pb1f2 does not interact with MAVS, why would it suppress signalosome recruitment?

line 338-344: PB1F2 was shown to be heavily ubiquitinated, H7PB1f2 also contains the C-terminal lysine cluster shown to be responsible for H5 and H1 PB1f2 expression level https://doi.org/10.1371/journal.pone.0118477. If H7PB1f2 does not interact with MAVS, authors should investigate if PB1f2 ubiquitination could contribute to MAVS ubiquitination by TRIM31

figure1. please comment on molecular size differences between wsn H5 and H5 PB1f2, analyze at least at the sequence level. The potential sites for posttranslational modifications like sumoylation, neddylation etc, those could be responsible for altered migration patterns as well as stability.

Fig.2b: in figure 1b presented data show no substantial difference between H7 PB1f2 regardless of MG132 presence while here fig2b difference is substantial. Please provide an explanation for results inconsistency and quantification with average expression level based on multiple independent repeats of this experiment.

**Part III – Minor Issues: Editorial and Data Presentation Modifications**

Reviewer #1: Figure 1 and 2 should be combined.

Reviewer #2: it would be beneficial if authors could perform co-staining with mitotracker and PB1f2, as pr8 PB1f2 was shown to efficiently co-localize with mitochondria regardless of plasmid transfection or viral infection

line 88-89 please provide appropriate reference

line 149-151: despite PR8 PB1f2 is most studied other variants including HK97 H5N1 PB1f2 were. https://doi.org/10.1371/journal.pone.0118477

please please edit accordingly

https://doi.org/10.1099/jgv.0.001220

please rephrase accordingly

line 240: please specify if co-expression was performed by co-transfection, or infection and transfection

line 287-305:

This section is hard to follow, does this mean that aggregation competent H7 PB1f2 interacts with MAVS before it aggregates, and at the steady-state expression of the MAVS, MAVS eventually aggregates making it resistant to H7PB1f2 interaction

Or am I missing something?

figure4.: To increase the relevance of luc-assays, please provide wb-data to demonstrate a relative level of PB1F2 along with a relative expression of RIG-IN, MyD88, MAVS TBK-1 IRF3-5D IFNB in the same sample (used for luciferase measurement).

figure 9A, B please include PB1-F2 portion of the wb around 12kDa as it has also Flag-taq

TRAF3 has Flag label in front of it while IKK and TKB1 behind it, please unify

PLOS authors have the option to publish the peer review history of their article (what does this mean?). If published, this will include your full peer review and any attached files.

Reviewer #1: No

Reviewer #2: No
---

## [Editor Report · Decision Letter 1]

7 May 2020

Dear Professor Jin,

We are pleased to inform you that your manuscript 'Virus subtype-specific suppression of MAVS aggregation and activation by PB1-F2 protein of influenza A (H7N9) virus' has been provisionally accepted for publication in PLOS Pathogens.

Best regards,

Peter Palese

Associate Editor

PLOS Pathogens

Ron Fouchier

Section Editor

PLOS Pathogens

Kasturi Haldar

Editor-in-Chief

PLOS Pathogens

orcid.org/0000-0001-5065-158X

Michael Malim

Editor-in-Chief

PLOS Pathogens

orcid.org/0000-0002-7699-2064
---

## [Editor Report · Acceptance letter]

2 Jun 2020

Dear Professor Jin,

We are delighted to inform you that your manuscript, "Virus subtype-specific suppression of MAVS aggregation and activation by PB1-F2 protein of influenza A (H7N9) virus," has been formally accepted for publication in PLOS Pathogens.

Best regards,

Kasturi Haldar

Editor-in-Chief

PLOS Pathogens

orcid.org/0000-0001-5065-158X

Michael Malim

Editor-in-Chief

PLOS Pathogens

orcid.org/0000-0002-7699-2064